



# Teleseismic P-waves at the AlpArray seismic network: Wave fronts, absolute traveltimes and traveltime residuals

Marcel Heinz Paffrath[1], Wolfgang Friederich[1], and the AlpArray and AlpArray-SWATH D Working Group [*]

[1]Ruhr-Universität Bochum
[*]Full list of participants at the end of the paper.

**Correspondence:** Marcel Heinz Paffrath (marcel.paffrath@rub.de)

**Abstract.** We present an extensive dataset of highly accurate absolute traveltimes and traveltime residuals of teleseismic P-waves recorded by the AlpArray Seismic Network and complementary field experiments in the years from 2015 to 2019. The dataset is intended to serve as the basis for teleseismic delay time tomography of the upper mantle below the greater Alpine region. In addition, the data may be used as constraints in full-waveform inversion of AlpArray recordings. The dataset com-
prises about 170.000 onsets derived from records filtered to $0.5\,\mathrm{Hz}$ and 214.000 onsets from records filtered to $0.1\,\mathrm{Hz}$. The high accuracy of absolute and residual traveltimes was obtained by applying a specially designed combination of automatic picking, waveform cross-correlation and beam-forming. Taking traveltime data for individual events, we are able to visualize in detail the wave fronts of teleseismic P-waves as they propagate across AlpArray. Variations of distances between isochrons indicate structural perturbations in the mantle below. Traveltime residuals for individual events exhibit spatially coherent patterns that prove to be reproducible if events of similar epicentral distance and azimuth are considered. When residuals for all available
events are stacked, conspicuous areas of negative residuals emerge that already indicate the lateral location of subducting slabs beneath the Apennines and the western, central and eastern Alps. Stacking residuals for events from 90 degree wide azimuthal sectors results in lateral distributions of negative and positive residuals that are generally consistent but differ in detail due to the differing direction of illumination of mantle structures by the incident P-waves. Comparing traveltimes from the $0.5\,\mathrm{Hz}$ and the $0.1\,\mathrm{Hz}$ dataset, we observe on average earlier arrivals in the $0.5\,\mathrm{Hz}$ dataset presumably caused by velocity dispersion.
Uncertainties of traveltime residuals are estimated from the peak width of the cross-correlation function and its maximum value. The median uncertainty is $0.15\,\mathrm{s}$ at $0.5\,\mathrm{Hz}$ and $0.18\,\mathrm{s}$ at $0.1\,\mathrm{Hz}$, way below the typical traveltime residuals of up to $\pm 2\,\mathrm{s}$. Uncertainties display a regional dependence caused by quality differences between temporary and permanent stations as well as location dependent noise conditions.

# 1   Introduction

The recently acquired AlpArray data set provides a fascinating opportunity to extend our knowledge on the structure of the upper mantle below the greater Alpine area, and in particular to answer long-standing questions regarding the orientation and penetration of lithospheric slabs, their connection to the well-studied crustal structure and their influence on surface processes.





AlpArray (Hetényi et al., 2018) is a multinational consortium built from 36 institutions from 11 countries dedicated to research
on the greater Alpine orogenic system encompassing the Alps, the Apennines, the Carpathians and the Dinarides. At its core
is the AlpArray Seismic Network, consisting of up to 600 seismic broadband stations operated in changing configurations
since 2015. With the Alps at its centre, the array reaches from the Po plain to the river Main, and from the Massif Central
to the Pannonian basin. The array is constructed on a foundation of permanent stations with temporary stations deployed to
fill gaps and thus produce a rather regular array with about 50 km station spacing. In addition, complementary targeted array
experiments were carried out: ocean bottom seismometers were deployed in the Ligurian Sea and even denser subarrays were
installed in the southern Central and Eastern Alps (Heit et al., 2017) and along the $13.4°$E meridian (EASI, 2014).

To tackle the challenging research opportunities offered by the AlpArray data with regard to Alpine mantle structure, delay-
time tomography of teleseismic body waves certainly belongs to the methods of choice. In teleseismic tomography, the variation
of arrival times of body waves from distant earthquakes across the array are inverted for velocity perturbations below the array.
Models obtained with this technique using regional arrays are typically confined to the upper mantle. For the AlpArray Seismic
network the lower bound is around 500 km depth. Lateral resolution is limited by the station spacing of the array. The method
is mainly sensitive to volumetric perturbations of seismic velocity and does not give constraints on the location of internal
discontinuities. It has been used in many studies on mantle structure, for example Koulakov et al. (2002); Lippitsch et al.
(2003); Piromallo and Morelli (2003).
A method which reaches beyond teleseismic tomography is full waveform inversion (FWI) where entire or partial waveforms
are inverted for velocity and also density perturbations (e.g., Mora, 1987; Tromp et al., 2005; Fichtner et al., 2009; Butzer et al.,
2013; Schumacher et al., 2016). Predictions of waveforms for given velocity models are obtained by full 3D numerical forward
modelling making the method very expensive with regard to storage requirements and computation time. When applied to
teleseismic body waves, hybrid approaches are invoked to make the method numerically tractable (e.g., Monteiller et al., 2013;
Tong et al., 2014a, b): full 3D forward modelling is only done in a regional box below the array while wave propagation from
the distant earthquake to this box is done by less expensive methods which however assume laterally homogeneous or axially
symmetric earth structure.

One basic preparatory step for both methods is the determination of traveltimes. While the need of traveltimes is obvious
for delay time tomography, also teleseismic full waveform inversion can benefit from traveltimes. Since the waveforms are
typically band-passed to some (narrow) frequency range, they become oscillatory and waveform matching may suffer from
cycle skipping. In such a situation, absolute traveltimes as additional constraints can help to make waveform matching less
ambiguous. Traditionally, arrival times were determined by manual reading of onset times from seismic records, but it is well-
known that even manual readings are affected by different reading styles of analysts (e.g., Douglas et al., 1997; Diehl et al.,
2009b) and, hence, may suffer from substantial inconsistencies. Moreover, manual reading of hundreds of thousands of records
would require a forbidding amount of human effort. To cope with the ever increasing number of available seismic stations,
automatic procedures have been developed to determine arrival times.

One of the first automatic picking procedures that is still used as a fast signal detection method was introduced by Allen
(1978, 1982). It is based on a characteristic function (CF) which is calculated as the ratio of the average of a signal within





a short time window to that in a long time window (STA/LTA). The CF rises as soon as a signal with a higher amplitude

than the preceding noise is encountered in the short time average window. Baer and Kradolfer (1987) developed an automatic phase picker by modifying Allen's characteristic function and implementing a dynamic threshold. The algorithm developed by Küperkoch et al. (2010) modifies and applies the scheme of Saragiotis et al. (2002). Kurtosis or skewness of a seismogram is calculated in a moving window and the Akaike Information Criterion (Akaike, 1974) is applied to the resulting CF.

These approaches work well in the context of earthquake location but should be considered with caution in conjunction

with delay time tomography and teleseismic full waveform inversion where little traveltime differences between neighbouring stations matter. Especially in view of the newly available dense seismic arrays and our quest for ever improving spatial resolution of tomographic models, the accuracy of traveltime measurements plays an increasingly important role. To avoid errors in traveltime differences, correlation techniques have been developed where selected wave packets of two different records of similar waveforms are correlated to determine their relative time shift. VanDecar and Crosson (1990) developed a multi-channel

cross-correlation technique (MCCC), to receive high precision relative arrival times, by correlating each trace with every other. This method was also used in a recent study by Zhao et al. (2016), where a finite-frequency kernel method was used for a tomography of the central European subsurface. However, this method does not produce absolute arrival times, which are a prerequisite for the stabilisation of the FWI.

In this paper, we confirm that even advanced techniques of automatic reading of arrival times do not reach the accuracy

required by teleseismic delay time tomography on dense arrays. We demonstrate, using AlpArray data, that an appropriate combination of automatic picking, correlation measurements and beamforming can attain the required accuracy and provide both reliable traveltime residuals and absolute traveltimes. Applying this technique, we are able to map the propagation of P-wave fronts across the AlpArray network and to obtain sufficiently accurate traveltime residuals at all stations of the network. By analysing records of hundreds of teleseismic earthquakes, we can show the coherency and reproducibility of the residuals

and study their dependency on event azimuth and frequency. Stacking of event-specific traveltime residuals yields very stable patterns that already show, without doing any inversion, where high- and low velocity anomalies have to be expected in the mantle below. We shall use these time measurements in a later study for performing a teleseismic tomography and full waveform inversion.

## 2  Data Basis

Deployment of the main AlpArray backbone network Z3 was started in 2015 with the aim to close gaps in the existing permanent networks for a recording period of at least three years by installing over 280 temporary broadband seismometers. Among these are 23 ocean bottom seismometers deployed by the LOBSTER project for a period of 8 months from June 2017 to February 2018 closing a large station gap in the Ligurian Sea. The earliest complementary experiment, partly included in our dataset, is the Eastern Alpine Seismic Investigation (EASI) project with 55 stations deployed on a north south profile at

13.4°E crossing the Alps from the northern Alpine foreland to the Adriatic Sea which recorded ground motion for more than a year until August 2015. The second complementary experiment SWATH-D was carried out for two years starting at the end of





2017, further increasing station density in a key area of the central and eastern Alps, directly above a Moho jump, a possible slab gap and slab polarity switch, thereby adding another 154 seismic broadband stations to our dataset. Finally, we extended the coverage of our database to the north and south by adding permanent stations in central Germany and northern Italy, thus

obtaining a total of 1025 different seismic broadband stations with recording times scattered through a period of over four and a half years between 2015 and the end of 2019, with a peak station coverage of more than 720 stations in late 2017.

## 2.1   Teleseismic Tomography Database

From the available data described above, we assembled records suitable for teleseismic tomography of 974 teleseismic earthquakes with origin times between January 2015 and July 2019 and moment magnitude $5.5$ or higher. They encompass wave-

forms of all stations available in a $5°$ radius around a central position in the Alps located at $46°$N and $11°$E. Out of these we evaluated mantle phases for 766 events, all within distances between $35°$ and $135°$ relative to the central position, leading to a minimum event distance of $30°$ for the closest and a maximum event distance of $140°$ for the farthest station. Information on location and moment tensor was taken from the Global CMT catalogue distributed by the Lamont-Doherty Earth Observatory (LDEO) of Columbia University (Dziewonski et al., 1981; Ekström et al., 2012).

We produced a high frequency dataset ($db_{0.5}$) using a bandpass filter between $0.03\,\text{Hz}$ and $0.5\,\text{Hz}$, which turned out to be perfectly suited for combining automatic picking and cross-correlation of land stations records. Since, however, oceanic microseismic noise is rather strong in this frequency band, cross-correlation of OBS records was only possible for very strong earthquakes. For this reason, we assembled a second low frequency dataset ($db_{0.1}$) with lowpass filter upper corner frequency of $0.1\,\text{Hz}$. In this way, most of the oceanic microseismic noise could be avoided, however at the expense of pick accuracy and

resolution of teleseismic tomography.

The distribution of earthquakes of both datasets relative to the Alps strongly varies with azimuth and epicentral distance. Fig. 1 shows the distribution of 370 evens that were ultimately picked for the high-frequency dataset. The majority of the recorded waves reach the Alps from a sector between north and east ($0°$ to $90°$) mainly originating from the Pacific Ring of Fire at epicentral distances between $80°$ and $90°$. A second concentration of sources in a sector between WSW and WNW with

azimuths between $230°$ and $290°$ is produced by earthquakes in the subduction zones of North and South America. Epicentral distances in this sector are more broadly distributed than in the NE sector. There is a remarkable lack of events in a sector between about $100°$ and $230°$ as well as in the opposite direction between $290°$ and $340°$. To obtain at least a few usable records from the poorly covered sectors long recording periods are essential. Statistical anomalies regarding event distribution in time as well as in moment magnitude are not visible, except for a preference of low magnitude events coming from the

southeast.

## 3   Automatic determination of absolute traveltimes, traveltime residuals and uncertainties

In the following part, we will examine the capability of characteristic function picking algorithms to resolve traveltime residuals with an accuracy required for traveltime tomography. We will show the most prominent difficulties and demonstrate how we



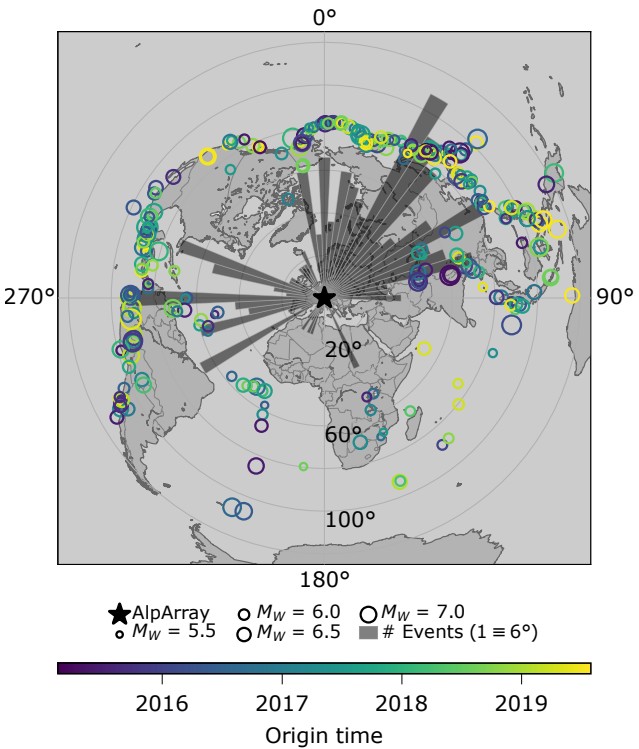

**Figure 1.** Event distribution of the high frequency dataset $db_{0.5}$. Size of circles correlates with moment magnitude, color with origin time. Histogram shows number of events binned in $5°$ bins azimuthally. A bar height of $60°$ radial distance equals 10 events coming from that direction. The distribution is very irregular with most events located in the northeastern quadrant and in a western sector. There are large gaps with few or no events especially from the southeast as well as from the northwest. Peak value is 18 events for the back-azimuth interval between $30°$ and $35°$.

can benefit from a combination of the AIC algorithm, beamforming and cross-correlation. The resulting multi-stage algorithm combines theoretical onset calculation for spherically symmetric earth models, characteristic function picking algorithms and various steps of signal cross correlation/beamforming to receive absolute as well as relative onsets with an uncertainty of fractions of a second. We also present an empiric way of automatic evaluation of uncertainties which has proven to be extremely robust.

### 3.1 Definitions and methodological approach

In the following, we will use the quantities absolute traveltime at some station, $\tau_j$, defined as the absolute arrival time minus source time, theoretical traveltime, $T_j$, defined as the time relative to the earthquake source time predicted by a standard earth model using an available earthquake location, and averages of these quantities over the entire array, $\overline{\tau}$ and $\overline{T}$, respectively. The





traveltime residual is defined by

$$r_j = \tau_j - T_j - (\overline{\tau} - \overline{T}) = \tau_j - \overline{\tau} - (T_j - \overline{T}). \tag{1}$$

We subtract array averages of observed and theoretical traveltimes to form residuals, because pure differences between observed and theoretical traveltimes contain errors of source time and depend on the wave path through the entire earth. The difference between the array averages, $\overline{\tau} - \overline{T}$, should take into account most of the heterogeneous earth structure remote from the array, while the remaining residuals after average subtraction should rather reflect influences of heterogeneities below the array.

The crucial question is, how to obtain highly accurate traveltime residuals and absolute traveltimes. If we were able to
construct a low-noise beam trace associated with some selected reference station by stacking appropriately shifted waveforms of all or selected stations on top of the reference station trace, we could perform cross-correlation with all other traces and read a highly accurate absolute traveltime from the beam trace. Let us denote this time by $\tau_B$. The theoretical traveltime for the beam is of course equal to that of the reference station, i.e. $T_B = T_R$. Taking the difference of the traveltime residuals of some station trace, $r_j$, and the beam trace, $r_B$, we find using Eq. (1):

$$r_j - r_B = \tau_j - \tau_B - (T_j - T_B)$$
$$= \Delta\tau_{jB} - \Delta T_{jB} \quad \text{with} \quad \tau_j = \tau_B + \Delta\tau_{jB}, \tag{2}$$

and hence

$$r_j = r_B + \Delta\tau_{jB} - \Delta T_{jB} \quad \text{with} \quad r_B = \tau_B - \overline{\tau} - (T_B - \overline{T}). \tag{3}$$

We estimate $\Delta\tau_{jB}$ with high accuracy by cross-correlation of station and beam trace and obtain $\tau_B$ by automatic picking.
From these two, we can calculate a highly accurate absolute traveltime for any station, $\tau_j = \tau_B + \Delta\tau_{jB}$, and the array average $\overline{\tau} = \tau_B + \overline{\Delta\tau_{jB}}$, finally allowing us to determine the beam residual, $r_B$, and the station residual $r_j$. Note that the error of the beam traveltime does not propagate into the beam residual, as it is also contained in the traveltime array average, $\overline{\tau}$, and subtracted when calculating $r_B$ according to Eq. (3). Thus, the accuracy of the residual is controlled by the accuracy of the $\Delta\tau_{jB}$ only.

To obtain the beam itself, we first select a reference station and consider traveltime differences to all other stations, $\tau_j - \tau_R$, which are again determined by cross-correlation. The reference station should be close to the center of the array to minimize waveform discrepancies to other stations, and exhibit a high data availability and low noise. We then use these time differences to shift the station traces and stack them on top of the reference trace to form the beam. Stacking is restricted to traces with sufficiently high correlation with the reference trace. To perform these initial cross correlations efficiently, we take advantage
of automatic readings at the stations based on higher-order statistics and the Akaike information criterion.

In summary, we start with automatic picks at the stations, use them to efficiently determine the time differences to a reference trace by cross-correlation, then shift the traces accordingly to form the beam. The beam is automatically picked and cross-correlation with all traces is repeated to obtain absolute traveltimes and traveltime residuals according to eqs. (2) and (3), respectively.





## 3.2 Higher Order Statistics picking algorithm

To get initial P-wave onsets in records of teleseismic earthquakes we use the HOS/AIC algorithm by Küperkoch et al. (2010), which is originally designed for precise local to regional earthquake detection, location and focal mechanism estimation but not for teleseismic phase reading. Therefore, all wavelength dependent parameters were adapted to our needs.

We choose kurtosis, the central moment of order $k = 4$, as characteristic function, which is calculated on a demeaned seismogram in a moving window of $N$ time samples at index $j$ as

$$\hat{m}_k(j) = \frac{1}{N} \sum_{l=0}^{N-1} x_{j-l}^k. \tag{4}$$

The AIC, which estimates the information loss of a function, is applied to the kurtosis function in the following way (Küperkoch et al., 2010):

$$AIC(k) = (k-1) \lg \left( \frac{1}{k} \sum_{j=1}^{k} CF_j^2 \right)$$

$$+ (L - k + 1) \lg \left( \frac{1}{L - k + 1} \sum_{j=k}^{L} CF_j^2 \right), \tag{5}$$

with $L$ being the length of the kurtosis function and $k$ ranging from $0$ to $L$. The minimum of the AIC in the calculation window is defined as the most probable pick (mpp) of the phase.

As initial guess, we use theoretical onsets of the phase estimated for a spherically symmetric earth model and calculate characteristic functions in a properly chosen time window around those onsets. We select the moving time window a full order of magnitude larger than those typically used for local earthquake onset determination, rendering it rather a growing than a moving window and calculate the most probable onset. Subsequently, an automatic quality is assigned to the onset based on the signal-to-noise ratio and and the difference between the latest and earliest possible pick (Diehl et al., 2009b). This quality determines whether the pick is used for further processing. The earliest possible pick, $t_{epp}$, is calculated as half the signal period before the most probable pick, $t_{mpp}$, accounting for a possibly missed first oscillation before the most probable pick. The signal period for this step is estimated by the mean time differences of zero-crossings within a characteristic time window after the most probable pick. The latest possible pick, $t_{lpp}$, is set to the time where the signal amplitude exceeds the noise level which is calculated as the root mean square of the noise in a window preceding the most probable pick. A symmetrized pick error (SPE) is then calculated as a weighted average of both pick uncertainties with double weight on the uncertainty derived from the latest possible pick:

$$SPE = \frac{\Delta t_{earliest} + 2\Delta t_{latest}}{3}$$

$$= \frac{(t_{mpp} - t_{epp}) + 2(t_{lpp} - t_{mpp})}{3}$$

$$= \frac{2t_{lpp} - t_{epp} - t_{mpp}}{3}. \tag{6}$$



By definition, using a maximum frequency of $0.5\,\mathrm{Hz}$, we obtain a minimum uncertainty from the earliest possible pick of a full second. Assuming $\Delta t_{latest} = 0$, the minimum possible SPE will be $0.33\,\mathrm{s}$. However, more realistic uncertainties will likely range in the order of 1 to 2 seconds, which is already close to the maximum traveltime residuals expected from mantle
heterogeneities below the Alpine orogen. In many cases, pick uncertainties even exceed typical traveltime residuals of interest (Fig. 3a). In order to take full advantage of the high station density of AlpArray, it is therefore crucial to reduce the uncertainties of our onsets.

We manually validated that the large uncertainties result from difficulties of the characteristic function algorithm to find that part of the first P-wave onset which is similar in all traces. Reason is the relatively low amplitude of the P-onset which
is often overlain by different levels of station noise. The resulting most probable onsets therefore strongly jitter confirming estimated uncertainties of about one half of the signal period. Another downside of the characteristic function approach is the false picking of either later arriving phases due to the first motion being completely masked by noise, or of other signals produced in the vicinity of the station leading to a severe number of outliers and to a time-intensive manual postprocessing.

Despite being unusable for a direct estimation of teleseismic traveltime residuals, with their uncertainties of a few seconds,
AIC onsets are still far superior to theoretical 1D earth model onsets as anchor points for signal cross-correlation. We found that theoretical phase onsets can differ from actual arrivals by up to some tens of seconds, most probably owing to differences of the true physical parameters in the global earth to those of the spherically symmetric earth model, uncertainties in origin time, as well as dispersion processes along the travel path. The resulting need of large cross-correlation shifts to catch all overlapping phases would involve a high risk of cycle skipping.

An analysis of the necessary shift in traveltimes predicted by the standard earth model AK135 (Kennett et al., 1995) for the final picks of 370 events in a frequency band between $0.03\,\mathrm{Hz}$ and $0.5\,\mathrm{Hz}$ yielded an average value of $-3.71\,\mathrm{s}$, implying that the average traveltime in the area of study is less than predicted by the AK135 earth model. The standard deviation is $\sigma = 5.84\,\mathrm{s}$. We found an absolute time-shift of over $10\,\mathrm{s}$ for 22 events with a peak value of $-53\,\mathrm{s}$. Hence, it is not reasonable to directly use 1D theoretical onsets as starting points for a signal cross-correlation.

Especially for lower-magnitude events and high-noise OBS records it may happen for some stations that useful automatic picks are not available. Provided that there are sufficient records left with a reliable automatic pick, we go back to theoretical traveltimes as correlation anchor points which have been corrected by the median time difference between the available automatic picks and the corresponding theoretical traveltimes. In this way, we still obtain good anchor points for cross-correlation and avoid omitting all records with unreliable automatic picks. This approach can greatly increase the yield of the cross-
correlation technique.

## 3.3 Correlation Approach

Applying a cross-correlation method to improve first arrivals on a large regional array like the AlpArray seismic network foots on the hypothesis of a high similarity of the waveforms of the selected phase across the array. We found this requirement to be satisfied especially well for teleseismic P-waves travelling through the mantle but not for PKP phases that penetrate the core.



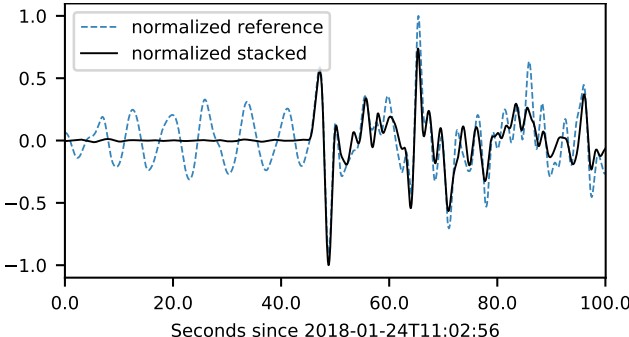

**Figure 2.** Stacking example for M6.1 earthquake in Hokkaido, Japan on 24 January 2018. 450 out of a total of 746 stations with a cross-correlation coefficient $> 0.8$ to the reference trace CH.PANIX (blue line) are stacked onto each other (black line). The first motion that was poorly resolvable on the reference trace can be clearly identified on the stacked trace, as the signal-to-noise ratio highly increased in the stacking process. Both traces are normalized for comparison and filtered between $0.03\,\text{Hz}$ and $0.5\,\text{Hz}$.

In contrast to mantle P-phases, PKP phases are composed of several arrivals which modify the shape of the waveform across the array owing to the different epicentral distances making signal correlation challenging.

We start by searching for a reference station which represents the waveforms of the entire array best for each single event. The most important criterion for such a station is a continuous operation with a high data quality. Therefore, we only consider permanent stations with low noise that were ideally running for the entire time span of events in our database. Also we want

this station to be in a central position in the Alps, with a low distance to all other stations it shall represent, to minimize possible changes in waveform related to large scale heterogeneities in the global earth (see Sect. 3.1). For each event we start with a small pool of stations meeting those criteria and correlate the signals of all other stations in small time windows around the anchor points we get from the AIC picks and the corrected theoretical traveltimes. The reference station with the highest mean correlation is then chosen as representer of the full set of stations for this event. Combining each station selected for

an event with all other available stations leads to 187.000 correlation pairs for the 370 events in our database. The average cross-correlation coefficient for those signal pairs is $0.78$.

After correlating all stations with the reference station, we align the waveforms according to the time of the maximum of the cross-correlation function. For each event we then form a beam representing the onset of the first P-wave phase by stacking the vertical component traces onto the reference station if the maximum cross-correlation coefficient is above a certain threshold.

For this study, we chose a threshold of $0.8$. To find the exact time of highest correlation independent from sampling, a parabola is fitted around the concave part of the cross-correlation function and analytically evaluated for its apex (Deichmann and Garcia-Fernandez, 1992).

The resulting beam (Fig. 2) is of very high quality with a drastically higher signal-to-noise ratio (SNR). On the beam, first motion becomes clearly visible and can be determined precisely using the automatic picking algorithm or by hand once for

each event. After determination of the absolute pick on the beam, vertical components of all stations are correlated with it for



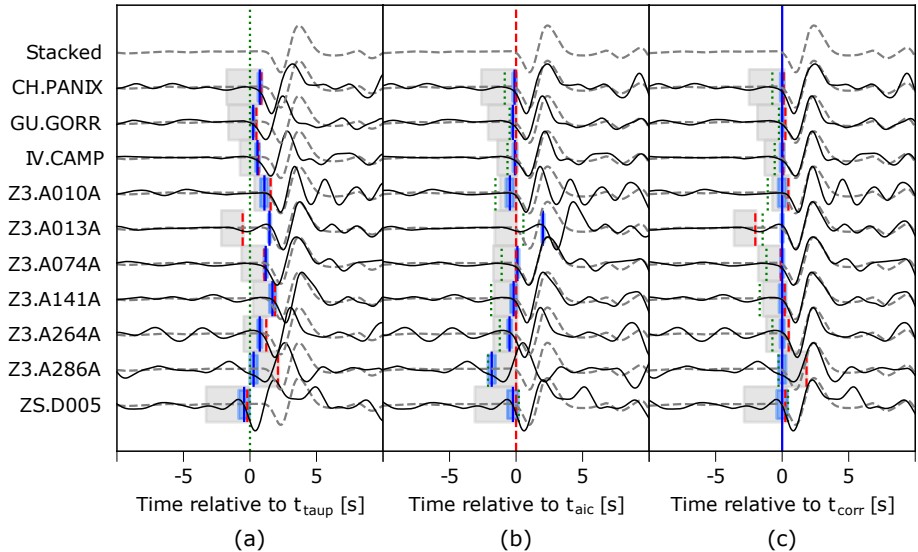

**Figure 3.** Waveform fit for P-arrivals of a M5.6 event on 7 August 2018 using the beam waveform as reference (dashed line). Left panel: Traces of different exemplary stations aligned by their theoretical onset (green dotted line). Travel time residuals (not demeaned) for each trace can be read from the differences of AIC (red dashed line) and correlation onsets (blue solid line) to the theoretical onset. Onset uncertainties are displayed by shaded areas in grey and blue colors, respectively. There is a good agreement between AIC and correlation onsets, however the estimated uncertainty of the AIC onsets is large, often exceeding the residual to the theoretical onset. Middle panel: Alignment of traces by their AIC onset. Overlap with the beam trace is good, but fails in certain cases of higher noise, which can lead to too early (e.g. Z3.A013A) as well as too late (e.g. Z3.A286A) AIC onsets. Right panel: Alignment by the correlation corrected onsets. Overlap with the beam trace is close to ideal. The estimated uncertainty of correlation corrected onsets is by a factor of about 10 lower than that of the AIC picks. Note the increased uncertainty for trace ZS.D005 and Z3.A010A exhibiting significant coda.

a second time. The traveltime of each station is then calculated as the beam traveltime plus the time difference to the beam traveltime which is obtained from the lag time associated with the maximum correlation.

The different role of theoretical, AIC and correlation corrected traveltime is illustrated in Fig. 3. If the traces are aligned according to theoretical traveltime (Fig. 3a), the alignment with the beam trace is worst. Evidently, this must be due to lateral heterogeneities below the array not contained in the standard earth model. If the traces are aligned according to their AIC automatic pick (Fig. 3b), overlap with the beam trace improves but there are still significant deviations for example for stations Z3.A013A and Z3.A286A. By construction, the agreement with the stacked trace is best when the traces are aligned according to their correlation corrected onsets (Fig. 3c). This latter subfigure demonstrates the jitter of the AIC picks which makes them unsuitable for teleseismic tomography.






### 3.4 Error estimation

Estimating an error for automatically determined as well as for manually assigned traveltimes is a difficult task and can be rather subjective. The concept of earliest and latest possible pick for error estimation uses information of a single trace only and is not suited for traveltime residuals determined by cross-correlation as the credibility of a time difference to a reference trace associated with a high cross-correlation coefficient is by far higher. This argument weighs even stronger, if the reference trace is a low-noise beam where the concept of estimating the earliest possible pick as half the signal wavelength is even more questionable as the first onset may be clearly identifiable without any risk to miss the first oscillation.

As the beam represents the waveform of the majority of stations, we consider the maximum cross-correlation between station and reference trace as the most important indicator for the relative accuracy of a traveltime difference. However, this assumption only holds if the stations forming the beam trace are evenly distributed in the array and not just representing a part of the array (for example stations close to the reference station). This is vital for the consistency of the full dataset.

Moreover, using the cross-correlation coefficient as a measure of accuracy might lead to a down-weighting of traces of stations influenced by strong local heterogeneities whose waveform does not fit the shape of the reference trace. Fortunately, this matter can be easily identified by looking at spatial distributions of maximum correlation. Affected stations should stand out in comparison to adjacent stations when looking at correlation coefficients averaged over many events (Sect. 5). We tried to find such regional dependencies by creating spatial plots of the cross-correlation coefficient for randomly selected events, but could not find any signs for a decrease in correlation coefficient with distance to a reference station, or any regional cluster of high or low cross-correlation coefficients.

A second criterion for a good match of station and reference trace is the shape of the cross-correlation function itself. Hence, we also evaluate the full width at half maximum (FWHM) of the cross-correlation function. If the FWHM increases, the cross-correlation maximum looses sharpness and the accuracy of a traveltime difference decreases. This approach implies a frequency dependency of traveltime uncertainty, leading to a higher uncertainty for longer periods (and hence wavelengths).

For a parabola fitted to the maximum of the cross-correlation function of the form:

$$f(x) = ax^2 + bx + c \tag{7}$$

the full width at half maximum (FWHM) can be calculated as

$$\text{FWHM} = 2\sqrt{\left(\frac{b}{2a}\right)^2 + \frac{C_{max} - 2c}{2a}}, \tag{8}$$

where $C_{max}$ denotes the maximum correlation. To combine both criteria, we chose to calculate the traveltime difference uncertainty as follows:

$$\sigma = (1 - C_{max})\,\text{FWHM} \tag{9}$$

The influence of a bad fit owing to signal coda on the cross-correlation coefficient and hence traveltime residual uncertainty is illustrated in Fig. 3c. The contribution of the width of the cross-correlation function, depending on signal period, is practically





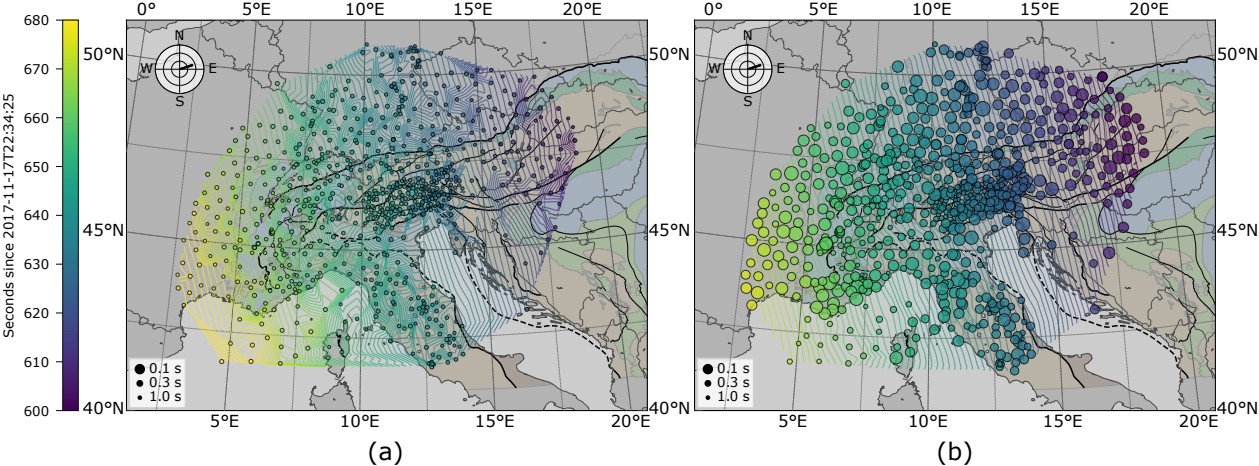

**Figure 4.** Traveltime fields of onsets from (a) the AIC algorithm that were only corrected for heavy outliers and (b) the combined cross-correlation AIC algorithm. Onset certainty increases with circle sizes. Isolines are linearly interpolated with temporal distances of 1 s.

the same for all traces of this event. However, the maximum correlation decreases for stations with significant coda (ZS.D005 and Z3.A010A).

## 4 Regional variation of traveltimes and traveltime residuals

In the following, we examine the variation of traveltimes and traveltime residuals across the array, study their dependence on
event azimuth and in particular delve into the reproducibility and consistency of the traveltime residuals. Especially, the latter is a crucial prerequisite for a successful tomographic inversion.

### 4.1 Wave fronts and spatial patterns of traveltime residuals

We start with teleseismic P-wave fronts constructed as isolines from the estimated traveltimes. To further stress the superiority of correlation-corrected traveltimes over AIC traveltimes, we show interpolated P-wave fronts constructed from both kind of
traveltimes. As an example, we take the M6.5 earthquake that happened on 17 November 2017 in the Eastern Xizang-India Border Region (Fig. 4). In both cases, one can identify the P-wave traveling across the array of about 700 stations from northwest to southeast. However, when constructed from the AIC onsets (Fig. 4a), the wave fronts are strongly irregular and several outliers are apparent leading to distorted isolines which cannot be explained by mantle heterogeneities. After application of the cross-correlation correction, the resulting wave fronts do no longer show the jitter inherent to the AIC onsets and become
smooth except for some weak undulations (Fig. 4b). These seem to be produced by several adjacent stations and should be attributed to subsurface structures.

To illustrate the varying shapes of the wave fronts crossing the AlpArray network from different azimuths and epicentral distances, we have selected four different earthquakes as representative examples: two with nearly equal back-azimuth (75°) but



**Figure 5.** Wave fronts and traveltime residual patterns of different earthquakes. Left panel: Absolute traveltimes, distance of isolines $1\,\mathrm{s}$, circle sizes inversely proportional to pick uncertainty. Right panel: Demeaned traveltime residuals relative to 1D earth model. (a), (b): M6.6 event, 2017-05-29, Sulawesi, Indonesia, BAZ=$76°$, distance=$104°$; (c), (d): M6.6 event, 2016-11-25, Tajikistan-Xinjiang Border Region, BAZ=$75°$, distance=$45°$; *Continued figure:* (e), (f): M6.6 event, 2017-08-18, North of Ascension Island, BAZ=$288°$, distance=$89°$; (g), (h): M6.8 event, 2017-06-22, Near Coast of Guatemala, BAZ=$213°$, distance=$52°$;

very different epicentral distances ($104°$ and $45°$) and two others covering western ($288°$) and southern ($218°$) back-azimuths

with differing epicentral distances ($89°$ and $52°$) (Fig. 5). In addition to the P wave fronts, we show the demeaned traveltime residuals associated with each particular event. They should correlate with the wave fronts as deformations of the wave front should lead to traveltime residuals and vice versa. To compensate influences of different station elevations, we apply a constant traveltime correction on all residuals shown, assuming vertical propagation and a surface P-velocity of $5.5\,\mathrm{km\,s^{-1}}$.





**Figure 5.** Continued.

Comparing Fig. 5a and Fig. 5c reveals a striking difference of the 1 s traveltime isoline spacing which is much greater for
the distant event. This reflects the very different horizontal apparent velocity of the two wave fields which is controlled by
epicentral distance and is much higher for the more distant event. While the wave fronts are generally regular and smooth,
some strong distortions become visible in some places. For example, in Fig. 5a, there is a widening and distortion of the
isolines in northern Italy north and east of the Ligurian Sea. The distortion and advancement can be associated with very large
negative residuals beginning at about 7.5°E and 45°N and continuing further to the northeast. The widening can be explained
by the transition from normal to large negative residuals further to the southwest. A second one occurs in the Apennines to
the south where the wave front has a strong lag near the western coast of Italy compared to the areas north of it but takes up





again while propagating over the areas with negative residuals in the western and central Apennines. In Fig. 5c, a very similar behaviour is visible.

A closer look at the traveltime residuals, Fig. 5b and Fig. 5d, reveals that there is a general agreement between the patterns but

also significant differences, for example, in south-eastern France where we observe normal to negative residuals for the distant event but positive residuals for the close event. The opposite is the case in most of Switzerland where we observe negative residuals for the close event and rather normal residuals for the distant one. Apparently, the steeply upwards propagating waves from the distant event see different mantle structures than the more slanted waves of the close event do.

A comparable behaviour is observed for events arriving from other back-azimuths. Isoline spacing is again much larger for

the more distant event whose waves arrive from a WNW direction. In Fig. 5g, there are again strong distortions of the wave fronts around $7.5°$E and $45°$N. These distortions are shifted to the NE for the waves arriving from the SSW in Fig. 5e. The associated residuals exhibit large-scale coherent patterns of negative and positive residuals but are again different in various regions. For example, residuals are generally positive in south-eastern France for the waves arriving from SSW while they are normal to negative for the event from WNW. This is again an indication of heterogeneous mantle structure to be resolved by

tomography later.

## 4.2   Stacked residuals

Although traveltime residuals differ with epicentral distance and event back-azimuth as waves move through high or low velocity zones from different angles before reaching the surface, there are certain features which tend to occur for a large number of events. The most prominent ones are the negative residuals along the Apenninic and Alpine chain. We stacked

residuals for all analysed events to find out regions of stable negative or positive traveltime residuals. It is highly important to understand that after stacking of the demeaned traveltime residuals, the resulting residuals are relative to an unknown one-dimensional earth model defined by all events used for stacking and not to the standard earth model used to calculate traveltime differences in the first place (e.g., Aki et al., 1977). Hence, negative or positive residuals indicate higher or lower velocities, respectively, compared to this average model and not compared to a standard earth model.

As we only consider mantle events between $35°$ and $135°$ distance, the incidence angle differs by a maximum of only $\sim 13°$ in a 1D earth model (ak135). Hence, we anticipate that the major variation of event-specific residual patterns is due to the event back-azimuth and expect the influence of the incident angle on those patterns to be small in comparison. We have already noticed this when examining the different individual events (Fig. 5).

As the azimuthal distribution of the events in our database is strongly uneven (Sect. 2.1), it is important to balance out

the influence of events from different directions when stacking. Otherwise, the influence of back-azimuths with high event density (e.g. NE in Fig. 1) on traveltime residuals would completely overwhelm any effects we get from poorly covered directions. Hence, we create $30°$ wide back-azimuth bins and stack traveltime residuals at each station for all events reaching the station from that direction. The value of $30°$ was chosen as a good compromise between angular resolution and smooth event distribution. The distribution of available measurements for different back-azimuth bin sizes can be found in the supplementary

material (Fig. A1).





**Figure 6.** Stacked traveltime differences for 370 events of the high frequency dataset $db_{0.5}$. Circle sizes correlate with number of back-azimuth bins for each station (maximum = 12). Blue colors indicate structures $v_p$ values higher than average, red colors indicate $v_p$ values lower than average. Travel times are binned calculating the mean traveltime for all events within $30°$ bins to balance out directional influences. A traveltime correction is applied for the vertical offset using a constant near-surface velocity estimate. High velocity anomalies contoured: W - Western Alps, C - Central Alps, E - Eastern Alps, A - Apennines, L - Ligurian Basin. Tectonic map of the Alpine chains compiled by M.R. Handy

The most striking features of the stacked traveltime residuals are the negative residuals following the Alpine arc from $45°$N, $7.5°$E to $46°$N, $14.5°$E (Fig. 6). We subdivide those anomalies into three major parts that can be related to presumed slab remnants in the upper mantle. The western negative anomaly (W) can be clearly differentiated from a large zone of positive



residuals to the west. It follows the Alpine mountain chain to the east, bending south towards the Po-plain. The central negative
anomaly C is attached to anomaly W in the south but follows a very different strike indicating a lateral change in mantle
structure. Defining a separate negative anomaly in the Eastern Alps (E) bending circular to the south towards the Dinarides is
not that obvious but substantiated by the observations in the even denser SWATH-D array (see inlet in Fig. 6) which indicate
a narrow gap of nearly zero residuals between C and E. This finding points to another discontinuous lateral change of mantle
structure below.

To make sure that the resulting residual patterns are mainly influenced by deeper mantle structures and not only by strong
crustal heterogeneities, we calculated synthetic traveltime residuals for the P-wave velocity model of the crust and uppermost
mantle by Diehl et al. (2009a) only (Fig. A2) using the same teleseismic source and receiver setup as for our observed data. The
resulting residual patterns do not match the ones we get from teleseismic wave fields at all. Also, the amplitudes of residuals
for purely crustal velocity perturbations are by a factor of two lower than the observed ones due to its restricted depth extent.
Hence, we can be sure that residual patterns we observe are dominated by mantle structures and are only modified by crustal
heterogeneities. Nevertheless, it is important to account for the influence of crustal structures in a teleseismic tomography.

### 4.3   Azimuthal dependence

We already showed traveltime residuals for individual wave fields. To give a more stable impression of the azimuthal variation
of the residuals, we stacked 3 neighbouring $30°$ averages to cover the four major azimuthal sectors NE, SE, SW and NW
(Fig. 7). All of them exhibit the negative residuals following the Apennine and the Alpine chain, the generally normal-to-
negative residuals in the northern foreland, the generally normal-to-positive residuals in south-eastern France and the Pannonian
basin. We have roughly sketched the positions of the major negative residuals that we already presented on all maps in Fig. 6
to investigate how the residuals move laterally when illuminated from different directions.

To draw some first conclusions on the possible location of structures forming the residual pattern, we imagine a planar wave
front moving through the subsurface from each of the four major azimuthal quadrants, creating an imprint of the traveltime
residuals it accumulated on its way to the surface. Due to wave refraction, the propagation direction of the wave fronts is almost
vertical close to the surface, whereas for greater depths the incidence angle increases. It follows that residual patterns that persist
for all four major azimuths point to structures located closer to the surface. Features that start to move when illuminated from
different directions suggest structures that are located at greater depths, as their imprint will be shifted laterally. If a feature in
the residual pattern can not be recovered at all from another direction it is likely to be related to a very deep anomaly that lies
outside the overlapping area of the ascending seismic waves. However, it is always possible that the effect of one anomaly is
removed, or reinforced by superimposition with another.

A good example for a structure with a laterally moving imprint is the Apenninic anomaly (A) that we find at the western
Italian coast penetrating into central Italy with a strike of $130°$. When illuminated from the northeast (Fig. 7a), the negative
traveltime residuals tend to move to the southwest and can even be tracked by the OBS off the Italian coast. Looking at the
same anomaly illuminated from the southwest (Fig. 7c), we see that its imprint has moved to the northeast and into the Adriatic





**Figure 7.** Traveltime residuals stacked for $90°$ quadrants NE, SE, SW, NW. For each quadrant the traveltime average of three possible $30°$ bins is shown. Circles of stations with incomplete coverage appear smaller; e.g. in (b): several directions are missing for OBS from SE direction, which has lowest event coverage. Dashed lines show outlines of the same negative residual anomalies as in Fig. 6. Arrows indicate a lateral movement of anomaly imprints caused by an illumination of waves of different azimuths.

Sea where we loose its track due to missing seismological stations. Illuminating the structure from perpendicular directions (Fig. 7b and d), the resulting pattern of negative residuals remains mostly the same.

Another remarkable difference to the stacked traveltime perturbation pattern from all azimuths can be seen in the imprints of structure (C) and (E) between $12°E$ and $15°E$. Both show very strong negative traveltime residuals that can not be separated into two parts when illuminated from the northeast (Fig 7a). These residuals do not appear with the same intensity for other





azimuths, which is an indication of a northeast dipping of the corresponding high-velocity anomalies. The reason is that waves incident from northeast may accumulate a large negative traveltime residual while waves incident from other directions acquire only minor changes in traveltime. Illuminated from the southwestern direction, we see that only the imprint of the western anomaly (C) survives. Looking at the same anomalies from the northwest, we see again the imprint of structure (C) but with a lower amplitude and a possible imprint of a deeper part of structure (E) that has moved to the southeast. The high velocity anomaly building (C) might therefore be dipping more vertically with less extension into the mantle, but with a stronger positive velocity perturbation. Illuminated from the southeast we see only little evidence of the northern part of structure (C) and none of structure (E).

The western Alpine anomaly (W) is mostly insensitive to azimuthal variations, indicating shallow high-velocity structure. We see some tendency of residuals to move to the northwest when illuminated from the southeast. When illuminated from the opposite direction the pattern faints, which however, might be a result of a compensation of the negative residuals by the overlaying low-velocity sediments in the Po-plain.

## 4.4 Frequency dependence

Up to now not much was said about the low-frequency dataset with maximum frequency $0.1\,\mathrm{Hz}$ which we assembled due to the high noise on the OBS records in the higher frequency band. As for the $0.5\,\mathrm{Hz}$-dataset, we determined absolute traveltimes and traveltime residuals using the techniques described above. We find that the obtained values for traveltime differ systematically. On average, over all stations and events, the difference amounts to $-0.8\,\mathrm{s}$, indicating that the waves are slightly faster and hence arriving earlier at $0.5\,\mathrm{Hz}$ than at $0.1\,\mathrm{Hz}$.

One probable reason for this finding is velocity dispersion caused by attenuation in the earth. According to Liu et al. (1976), the dispersion effect is given by

$$\beta = \frac{v(f_2) - v(f_1)}{v(f_1)} = \frac{1}{\pi Q} \ln\left(\frac{f_2}{f_1}\right) \tag{10}$$

where in our case $f_2/f_1 = 5$. Assuming an average traveltime of the P-wave of about $800\,\mathrm{s}$ corresponding to an epicentral distance of about $90°$, the relative traveltime change is $1.0$ permille. Inserting this value into eq. (10) leads to a quality factor $Q_p$ of 500. Using the relation (Forbriger and Friederich, 2005)

$$Q_p^{-1} = \left(1 - \frac{4v_s^2}{3v_p^2}\right) Q_\kappa^{-1} + \frac{4v_s^2}{3v_p^2} Q_\mu^{-1} \tag{11}$$

with S-wave velocity $v_s$, P-wave velocity $v_p$, bulk quality factor $Q_\kappa$ and shear quality factor $Q_\mu$, and setting $v_p = 10\,\mathrm{km/s}$, $v_s = 6\,\mathrm{km/s}$ and $Q_\kappa^{-1} = 0$ as representative values of the lower mantle, we find $Q_p \approx 2Q_\mu$. Thus, a $Q_p$ of 500 corresponds to a shear quality factor of 250 which compares well with the value of 312 given in PREM for the lower mantle. This result suggests that the average traveltime bias is caused by dispersion in the global earth.

Another phenomenon that could cause a traveltime difference between the high- and low-frequency datasets is the finite frequency effect (finite wavelength effect) which expresses the fact that long waves see heterogeneities differently than short waves do. But we expect this effect rather to be important for wave propagation through the regional heterogeneities below





the Alpine region. In addition, regional heterogeneities of attenuation below the Alps could also contribute to the traveltime
differences.

To illustrate the differences between both frequency bands, we directly show traveltime residuals determined for the $0.1\,\mathrm{Hz}$
dataset (Fig. 8a), and we form differences of the traveltime residuals, i.e. we subtract the values for the low-frequency dataset
from those of the high-frequency dataset (Fig. 8b).

Both are stacked again in $30°$ bins, similar to the high frequency dataset, to account for the uneven azimuthal distribution.
At first glance, there are no striking differences between residuals of $db_{0.5}$ (Fig. 6) and $db_{0.1}$ (Fig. 8a). Both show very similar
patterns of negative and positive residuals. However, negative and positive anomalies of the $0.1\,\mathrm{Hz}$ dataset seem less well-
defined, mainly due to the smoothing effect of the larger wavelengths. Moreover, overall amplitude of the anomalies is slightly
weaker at low frequency also due to the smoothing effect. Conspicuous examples for this behaviour are the area of zero to
slightly positive residuals between anomalies W and A in Fig. 6 which seems to be smeared out in Fig. 8a and anomalies C
and E in Fig. 6 which are less strongly expressed in the $0.1\,\mathrm{Hz}$ residuals.

The differences between the stacked residuals for the $0.5\,\mathrm{Hz}$ and the $0.1\,\mathrm{Hz}$ dataset (Fig. 8b) show coherent regions of
positive and negative disparity with negative ones in the region of anomalies C and E as well as the south-eastern part of A
and positive disparity in the area of the Po plain between anomalies W and A. There is another area of positive disparity north
of the $45°$ parallel between 5 and $7.5°$longitude. This area is to the west of anomaly W where positive residuals are slightly
stronger in the $0.1\,\mathrm{Hz}$ dataset. In the remaining regions, deviations are rather small with a weak tendency to negative values
especially in the Alpine foreland.

There is a massive increase in the total number of picks for the ocean bottom seismometers. The number of OBS picks for
all events increases from 421 to 1113 ($+164\,\%$), compared to a $25\,\%$ increase only for all stations. This might have a notable
effect on resolving structures below the Ligurian Sea when performing a teleseismic tomography. In the case of our stacked
traveltime residuals, we do not see a strong effect, because most of what we would see balances out by the azimuthal binning.

The traveltime residual differences (Fig. 8b) are generally small below $0.5\,\mathrm{s}$ and close to zero for most stations. We see
positive values in the south of the central Alps and negative values in the western and northern parts of the western and central
Alps, respectively. In addition, there are slight negative values along the SWATH-D array in the eastern Alps. Interpretation
of the pattern is difficult because it is hard to estimate the effect of wavelength smoothing without knowing the 3D structure
of the mantle. For heterogeneities of attenuation, we would expect low dispersion (positive values) in slab regions with little
attenuation, and high dispersion (negative values) in hot low-velocity areas. At least in the area of the southern rim of the
Central Alps where the residuals are very negative indicating the presence of slabs, we observe a trend towards positive
differences. Moreover, in the western Alps and their forelands where we find positive residuals we observe slightly negative
residual differences. The same applies to the positive residuals in the Pannonian basin where we also find a tendency to negative
residual differences. Thus, heterogeneities of attenuation may play a role here.



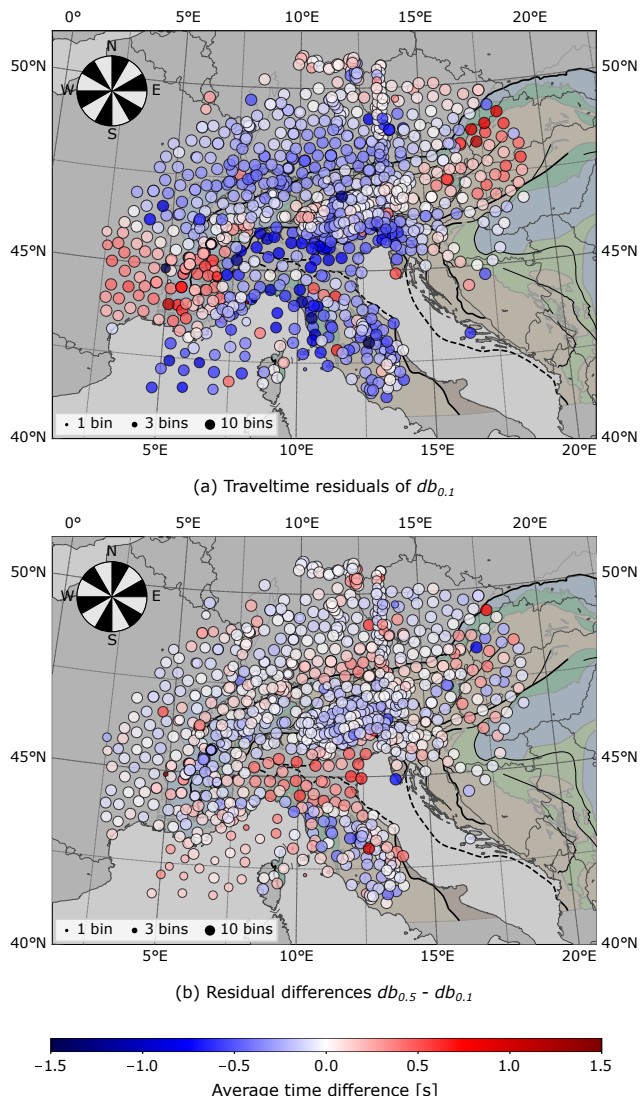

**Figure 8.** (a): Stacked traveltime residuals for $30°$ bins from all directions for $db_{0.1}$ similar to Fig. 6. Patterns look similar in most areas to the one for the high frequency dataset at first glance, except for weaker residual amplitudes. (b): Differences between stacked residuals of high and low frequency datasets show areas with positive disparity (e.g. Po-plain), or negative disparity (e.g. Apennines, Alpine arc). Proportion of pick differences (same ray available in each dataset) to total number of picks in $db_{0.5}$ is $96\%$.

## 5 Uncertainities of traveltime residuals

We categorize traveltime uncertainties into five different classes in steps of $0.1$ s ranging from class 0 (best) below $0.1$ s to class 4 (worst), over $0.4$ s. Comprising over 170.000 onsets, the traveltime uncertainty distribution of $db_{0.5}$ (Fig. 9a) has a maximum at $0.08$ s with about 46.000 ($27\%$) of the onsets in the highest class. Less than $10\%$ are in the lowest class of $0.4$ s or higher. The





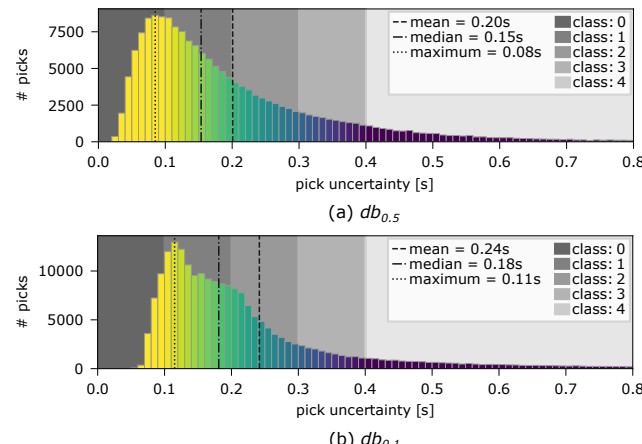

**Figure 9.** Pick uncertainty distribution of $db_{0.5}$ and $db_{0.1}$, clipped at $0.8\,\mathrm{s}$. Colormap of uncertainties is similar to Fig. 10

low-frequency dataset $db_{0.1}$ has an increased signal quality (i.e. higher SNR) (Fig. 9b) which is reflected in the higher number of picks (over 214.000), an increase of over $25\,\%$ compared to the high frequency dataset. However, the traveltime uncertainty distribution is drawn to higher values, with its mean being shifted by nearly half a class towards higher uncertainties. While the peak value of the uncertainty histogram is still in the same region as that of the high frequency dataset, there are only ca. $9.8\,\%$ of the total number of picks in class 0 and over $12\,\%$ in class 4. The reason for this counter-intuitive behaviour is the fact
that, owing to the greater signal periods, the maxima of the correlation function for estimating the time differences (Sect. 3.4) become wider leading to a higher error estimate.

## 5.1 Regional distribution

An evaluation of the regional distribution of the median of traveltime uncertainty per station in the $db_{0.5}$-dataset (Fig. 10a) exhibits lower values, north and east of the Alpine arc, in central and southern Germany as well as in the Czech Republic,
eastern Austria and Slovenia. We hypothesize that this effect originates in the spatial segregation of those areas from the Alpine orogen, as the subsurface structure of the surrounding area of the Alps is simple in comparison to that beneath the Alps. In contrast to that, traveltime uncertainty increases above the highly complex structures in the Alpine arc, where the P-wave fronts are significantly altered by the strongly heterogeneous subsurface. This decreases their correlation with the wave front signal on other stations of the array and herewith to the stacked reference trace. It is also likely that uncertainty increases due to
local site effects which can be significant in narrow valleys, where anthropogenic activities such as traffic are harder to evade. These influences should be visible on single stations which show a high daytime noise level. We expect those effects to be present equally in both, the high and the low frequency dataset. However, most of station outliers we see in one dataset is not present in the other.

The traveltime uncertainty distribution pattern of the lower frequency dataset $db_{0.1}$ (Fig. 10b) shows a comparable shape as
the high frequency one with the lowest uncertainty in the northeastern parts of the array. However, overall uncertainty is higher



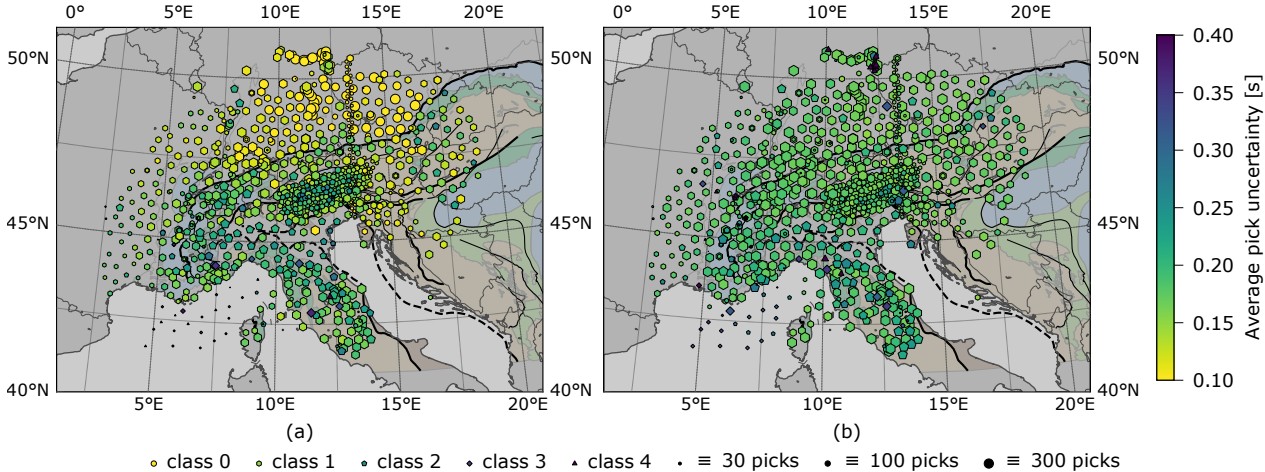

**Figure 10.** Maps of the median uncertainty of all picks for the high frequency (a) and the low frequency (b) dataset. Symbol sizes correlate with number of picks per station, color shows the average traveltime uncertainty and symbol shapes indicate the average pick class on each station. Traveltime uncertainty gets higher for $db_{0.5}$ in areas strongly influenced by deep subsurface structures, e.g. orogenic roots as well as strong heterogeneities close to the surface. Temporal coverage is best in the northern Alpine foreland, central Alps and Apennines. Complementary experiments are salient, as their measuring duration i.e. number of total picks is limited in contrast to other stations. The EASI experiment can be seen as a straight line of smaller sized symbols on a north-south directed profile, spatially (but with no temporal overlap) cutting through the SWATH-D experiment in the central Alps above the Tauern Window. The latter has a higher station density compared to the rest of the array. Ligurian Basin Ocean Bottom Seismometer (LOBSTER) are also clearly identifiable by a lower number of picks (smaller symbol size) as well as by a higher average uncertainty as a consequence of their noisy measuring environment.

and the contrast between regions of high and low uncertainty is decreased. We assume that this is an effect of signals of higher wavelengths being less sensitive to small scale anomalies due to their lower resolution capability (finite frequency effect) and hence, waveform fit with the reference trace is easier to achieve. The only area, where we see a totally opposed behaviour is the Ligurian Sea, where the positive impact on pick uncertainty using lower frequencies is salient. Here, not only the number of

total picks greatly increased, but also average pick quality is raised by a full class for nearly all OBS, whilst for the remaining stations quality tends to decrease by almost one class in comparison to the high frequency dataset. We also note that there are only small changes in uncertainty for the SWATH-D stations. They even show slightly counterintuitive behaviour, of having higher uncertainties than average in $db_{0.5}$, but lower uncertainties than average in $db_{0.1}$.

The total number of picks per station is highest on permanent station networks, which are distributed densest in the central

Alps and Apennines. Temporal coverage slightly decreases in the western part of the Array, due to a delayed start of deployment of temporary stations in this area.





## 6 Discussion

With large and dense arrays like AlpArray the amount of available records for traveltime measurements may readily accumulate to hundreds of thousands depending on the duration of the deployment. Hence, automatic procedures for determining traveltimes become mandatory. Moreover, with increasing resolution capabilities of such arrays, also demands on the accuracy of traveltimes have tightened, in particular if we want to resolve the correspondingly small travel time differences between nearby stations. Sophisticated automatic picking procedures apparently do not meet these demands. A way out of this dilemma are measurements of relative time shifts between two traces by cross-correlation. They can be automated and are particularly well suited for dense arrays which provide a wealth of similar waveforms. However, they do not provide absolute traveltimes. For this reason, stacking or beamforming to obtain stable low-noise reference traces is an essential further element in traveltime determination (Rawlinson and Kennett, 2004). Mitterbauer et al. (2011) already used such an approach in their teleseismic tomography of the eastern Alps even though they only determined about 6600 traveltimes. They stacked records aligned to automatic picks to obtain low-noise reference traces for ensuing cross-correlation measurements. After determining cross-correlation time-shifts they iterated the correlation and stacking step until a stable reference trace was obtained. Our approach is similar to that of Mitterbauer et al. (2011), as it also makes use of the elements automatic picking, beamforming and cross-correlation. These are absolutely essential to obtain the required accuracy. But we found that iterative correlation and stacking was not necessary with AlpArray data, neither for the $0.5\,\mathrm{Hz}$ nor the $0.1\,\mathrm{Hz}$ dataset. It proved sufficient to select one centrally located permanent station and correlate its waveform with the waveforms of all other stations to obtain time-shifts for constructing a stable and very-low noise reference or beam trace. Picking this beam trace automatically and using it as a reference trace for cross-correlation time-shift measurements was sufficient to obtain highly accurate relative and absolute traveltimes for up to 210.000 records in a fully automated fashion.

The uncertainty of a cross-correlation time delay measurement should evidently be related to the width of the maximum of the cross-correlation function where the time delay is read off but it remains a subjective choice in any case. We measure the full width at half maximum (FHWM) which is however a too conservative estimate of the real error. For this reason, we include the maximum normalised correlation $C_{max}$ as a second component into error estimation. The higher the maximum correlation the better is the delay time estimate. We reflect this expectation by scaling the FHWM by a factor of $1 - C_{max}$. This definition nicely includes the frequency dependence of uncertainty with a higher error at low frequencies because the signal gets smoother and onsets more emerging. In addition, it allows a consistent and automatic determination of uncertainty. Our reconstructions of smooth and nearly unperturbed wave fronts across the entire array from the observed travel time field with wave fronts separated by only 1 second of traveltime demonstrate that the estimated traveltime uncertainties of on average $0.2\,\mathrm{s}$ (median $0.15\,\mathrm{s}$) is realistic since otherwise conspicuous bumps should appear in the wave fronts, as they indeed do when the wave fronts are constructed from the automatic picks.

We do not carry out here a tomographic inversion of the dataset (which will be presented in a follow-up paper) but the maps of residual traveltimes already allow some inferences on the underlying mantle heterogeneities. Especially the maps of stacked residuals reflect robust spatial patterns of anomalies that occur in most of the event-specific residual traveltime maps. Most





remarkable are here the negative traveltime anomalies designated with the letters L, A, W, C and E in Fig. 6. The anomaly L in the Ligurian Sea apparently reflects thin oceanic crust underlain by high-reaching fast upper mantle leading to a reduction of travel times. The anomaly A strikes with the Apenninic mountain chain, forming a narrow band along the western Italian coast in the north and then opening up into a broad band below central Italy. It indicates slab-like high-velocity mantle material

beneath the Appennines. The slab seems to dip nearly vertically since the negative traveltime anomalies systematically move to the southwest (northeast) for waves arriving from the northeast (southwest) (Fig. 7). The anomaly W apparently reflects the generally southeast dipping subduction of European lithosphere in the Western Alps as previously inferred by Lippitsch et al. (2003). The lateral continuity of this anomaly ends at the transition from the Western to the Eastern Alps at about 10 degrees east where we find anomaly C which exhibits a very different strike. A further lateral transition to anomaly E seems to occur

at about 12 degrees east at the western rim of the Tauern window.

It is of course not possible to infer mantle structure just from the stacked traveltime residuals because they integrate over depth. This problem can only be solved by a tomographic inversion of the event-specific residual maps. Nevertheless, we can already recognize without any inversion that slab structure underneath the Alps is complicated and high-velocity structures do not form laterally coherent bodies along the Alpine chain. This laterally discontinuous behaviour of the traveltime residuals

matches previous findings by Lippitsch et al. (2003) and Mitterbauer et al. (2011) who identified different lateral slab segments in the western, central and eastern Alps with possible changes in slab dip. Especially, the slab structure belonging to anomaly (E) is disputed because Lippitsch et al. favor a north-dipping slab while Mitterbauer et al. infer a nearly vertical slab dip. An amplification of the negative traveltime residuals when illuminating the structure from northeast compared to other azimuths favors a northeastern dipping structure, as we could see in Fig. 7. Due to the complexity of the region, however, it is essential

to carry out a complete teleseismic tomography in order to obtain more precise information about the geometry.

Another interesting aspect of our traveltime measurements is their frequency dependence, in particular the differences between the traveltime residuals derived from the $0.5\,\mathrm{Hz}$ and the $0.1\,\mathrm{Hz}$ dataset. We argued with the smoothing effect of the larger wavelengths as the major cause of the discrepancies. However, dispersion through attenuation could also play a role. To analyse this contribution we go back to eq. (1) and write the differences between the residuals as

$$r_{j,0.5} - r_{j,0.1} = \tau_{j,0.5} - \tau_{j,0.1} - (\overline{\tau_{0.5}} - \overline{\tau_{0.1}})$$
$$- (T_{j,0.5} - \overline{T_{0.5}}) + (T_{j,0.1} - \overline{T_{0.1}}), \tag{12}$$

where the subscripts 0.5 and 0.1 denote the upper frequency limit of the dataset. Since we did not take into account dispersion in the theoretical traveltimes, their contribution cancels out and we are left with

$$r_{j,0.5} - r_{j,0.1} = \tau_{j,0.5} - \tau_{j,0.1} - (\overline{\tau_{0.5}} - \overline{\tau_{0.1}}). \tag{13}$$

If there was only global dispersion due to attenuation, we would get, according to eq. (10), $\tau_{j,0.5} - \tau_{j,0.1} = \beta\tau_{j,0.5}$ and $\overline{\tau_{0.5}} - \overline{\tau_{0.1}} = \beta\overline{\tau_{0.5}}$ leading to

$$r_{j,0.5} - r_{j,0.1} = \beta(\tau_{j,0.5} - \overline{\tau_{0.5}}). \tag{14}$$





As deviations of traveltime from the array average amount to less than $30\,\mathrm{s}$ and $\beta = 10^{-3}$, global dispersion should contribute less than $0.03\,\mathrm{s}$ to the observed frequency dependence of the residuals of up to $0.7\,\mathrm{s}$. Thus, global dispersion is not suited to explain the observed frequency dependence of the residuals. We can also try to estimate the possible size of the attenuation heterogeneity effect. Consider a slab of $300\,\mathrm{km}$ length within which the P-wave spends about $30\,\mathrm{s}$. We assume an average $Q_p$ in the upper mantle of about 200. If the slab's Q is 800, then there is a dispersion effect of

$$30\,s\,\frac{1}{\pi}\ln\frac{f_2}{f_1}\bigl(\frac{1}{200} - \frac{1}{800}\bigr) \approx 0.06\,\mathrm{s}\,, \tag{15}$$

which is also insufficient to explain the observed differences of up to $0.7\,\mathrm{s}$. We therefore conclude that the major contribution must stem from the finite frequency effect.

## 7 Conclusions

The dense AlpArray Seismic Network and its complementary deployments offer the unique opportunity to infer mantle structure beneath the greater Alpine region with an unprecedented resolution. However, to benefit fully from the array, absolute traveltimes and traveltime residuals of high accuracy and consistency are required. We have shown that even very sophisticated automatic picking algorithms based on higher-order statistics and the Akaike information criterium are unable to fulfill this requirement. We demonstrate that, instead, a hybrid approach combining characteristic function picking, waveform cross-correlation and beamforming techniques that takes advantage of the dense array is indeed capable of achieving the required accuracy. Since this hybrid approach is also fully automated, human effort is drastically reduced and the consistency of the generated dataset is ensured by the reproducibility of the automatically determined onsets. Beamforming requires similar waveforms posing demands on array density depending on frequency range. AlpArray proved to be sufficient for high correlation at the chosen lowpass filter frequencies ($0.5\,\mathrm{Hz}$ and $0.1\,\mathrm{Hz}$). Admitting higher frequencies may require smaller interstation distances to preserve waveform coherency.

The accuracy of traveltimes and residuals is validated by the fact that they allow a reliable and flawless construction of teleseismic wave fronts in terms of traveltime isochrons. These exhibit small undulations indicating the presence of mantle heterogeneities. The traveltime residuals for individual events show very coherent and reproducible spatial patterns that perfectly fit to these undulations and, although masked by their dependence on illumination incidence and azimuth, already give a glimpse on mantle velocity anomalies, in particular conspicuous slab-like high velocity structures along the Alpine arc and the Apennines. Studying the azimuthal variations of the residuals provides first hints on the dip of these anomalies. Even stacks of residuals maps from hundreds of events show distinct, spatially coherent areas of positive and negative residuals and, in particular, reproduce the conspicuous negative residuals. These results indicate the stable presence of mantle heterogeneities in each map of traveltime residuals and allow us to make assertions about the geometry and position of the high and low-velocity objects below the Alps even before performing a full teleseismic tomography.

Determining traveltimes and residuals from data filtered to different maximum frequencies reveals an average shift of nearly $1\,\mathrm{s}$ towards earlier travel times for the high frequency ($0.5\,\mathrm{Hz}$) dataset. We ascribe this finding to global dispersion caused by



attenuation in the earth. Maps of traveltime residuals for the $0.5\,\mathrm{Hz}$ and $0.1\,\mathrm{Hz}$ dataset show similar patterns but are different with respect to amplitude and sharpness of the anomalies confirming that the sensitivity of waves to heterogeneities depends on wavelength. Hence, datasets of traveltime and residuals obtained from differently filtered waveforms can not be used together in a classical traveltime tomography.





## Appendix A: Supplementary Material

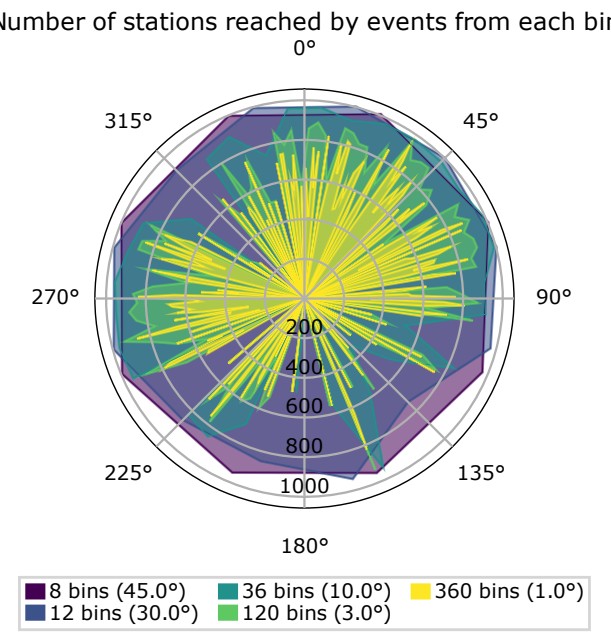

**Figure A1.** Number of stations for each bin for different bin sizes. Distribution is rather homogeneous for $30°$ and $45°$ bins. With smaller bin sizes bias increases, which downweights back-azimuths with low event coverage.

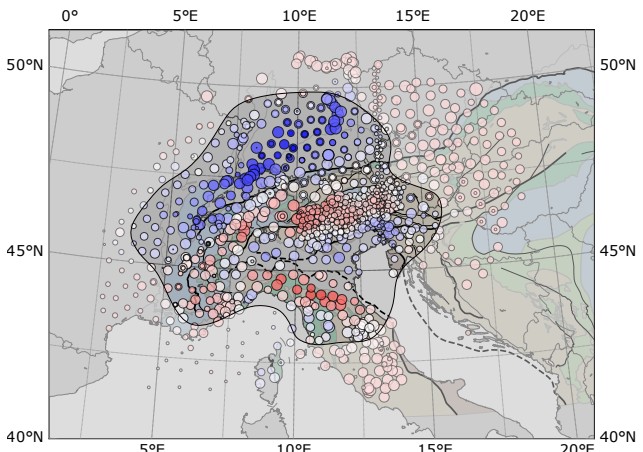

**Figure A2.** Demeaned and stacked traveltime residuals for synthetic test with near-surface model by Diehl et al. (2009a). Model boundaries are roughly outlined to show influence of heterogeneities on traveltime pattern.





*Team list.* The AlpArray Seismic Network team: György HETÉNYI, Rafael ABREU, Ivo ALLEGRETTI, Maria-Theresia APOLONER, Coralie AUBERT, Simon BESANÇON, Maxime BÈS DE BERC, Götz BOKELMANN, Didier BRUNEL, Marco CAPELLO, Martina ČARMAN, Adriano CAVALIERE, Jérôme CHÈZE, Claudio CHIARABBA, John CLINTON, Glenn COUGOULAT, Wayne C. CRAW-FORD, Luigia CRISTIANO, Tibor CZIFRA, Ezio D'ALEMA, Stefania DANESI, Romuald DANIEL, Anke DANNOWSKI, Iva DASO-VIĆ, Anne DESCHAMPS, Jean-Xavier DESSA, Cécile DOUBRE, Sven EGDORF, ETHZ-SED Electronics Lab, Tomislav FIKET, Kasper

FISCHER, Florian FUCHS, Sigward FUNKE, Domenico GIARDINI, Aladino GOVONI, Zoltán GRÁCZER, Gidera GRÖSCHL, Stefan HEIMERS, Ben HEIT, Davorka HERAK, Marijan HERAK, Johann HUBER, Dejan JARIĆ, Petr JEDLIČKA, Yan JIA, Hélène JUND, Edi KISSLING, Stefan KLINGEN, Bernhard KLOTZ, Petr KOLÍNSKÝ, Heidrun KOPP, Michael KORN, Josef KOTEK, Lothar KÜHNE, Krešo KUK, Dietrich LANGE, Jürgen LOOS, Sara LOVATI, Deny MALENGROS, Lucia MARGHERITI, Christophe MARON, Xavier MARTIN, Marco MASSA, Francesco MAZZARINI, Thomas MEIER, Laurent MÉTRAL, Irene MOLINARI, Milena MORETTI, Anna NARDI, Jurij

PAHOR, Anne PAUL, Catherine PÉQUEGNAT, Daniel PETERSEN, Damiano PESARESI, Davide PICCININI, Claudia PIROMALLO, Thomas PLENEFISCH, Jaroslava PLOMEROVÁ, Silvia PONDRELLI, Snježan PREVOLNIK, Roman RACINE, Marc RÉGNIER, Miriam REISS, Joachim RITTER, Georg RÜMPKER, Simone SALIMBENI, Marco SANTULIN, Werner SCHERER, Sven SCHIPPKUS, Detlef SCHULTE-KORTNACK, Vesna ŠIPKA, Stefano SOLARINO, Daniele SPALLAROSSA, Kathrin SPIEKER, Josip STIPČEVIĆ, Angelo STROLLO, Bálint SÜLE, Gyöngyvér SZANYI, Eszter SZŰCS, Christine THOMAS, Martin THORWART, Frederik TILMANN, Stefan

UEDING, Massimiliano VALLOCCHIA, Luděk VECSEY, René VOIGT, Joachim WASSERMANN, Zoltán WÉBER, Christian WEIDLE, Viktor WESZTERGOM, Gauthier WEYLAND, Stefan WIEMER, Felix WOLF, David WOLYNIEC, Thomas ZIEKE, Mladen ŽIVČIĆ, Helena ŽLEBČÍKOVÁ. The AlpArray SWATH-D field team: Luigia Cristiano (Freie Universität Berlin, Helmholtz-Zentrum Potsdam Deutsches GeoForschungsZentrum (GFZ), Peter Pilz, Camilla Cattania, Francesco Maccaferri, Angelo Strollo, Günter Asch, Peter Wigger, James Mechie, Karl Otto, Patricia Ritter, Djamil Al-Halbouni, Alexandra Mauerberger, Ariane Siebert, Leonard Grabow, Susanne Hemmleb, Xi-

aohui Yuan, Thomas Zieke, Martin Haxter, Karl-Heinz Jaeckel, Christoph Sens-Schonfelder (GFZ), Michael Weber, Ludwig Kuhn, Florian Dorgerloh, Stefan Mauerberger, Jan Seidemann (Universität Potsdam), Frederik Tilmann, Rens Hofman (Freie Universität Berlin), Yan Jia, Nikolaus Horn, Helmut Hausmann, Stefan Weginger, Anton Vogelmann (Austria: Zentralanstalt für Meteorologie und Geodynamik (ZAMG)), Claudio Carraro, Corrado Morelli (Südtirol/Bozen: Amt für Geologie und Baustoffprüfung), Günther Walcher, Martin Pernter, Markus Rauch (Civil Protection Bozen), Damiano Pesaresi, Giorgio Duri, Michele Bertoni, Paolo Fabris (Istituto Nazionale di Oceanografia

e di Geofisica Sperimentale OGS (CRS Udine)), Andrea Franceschini, Mauro Zambotto, Luca Froner, Marco Garbin (also OGS) (Ufficio Studi Sismici e Geotecnici -Trento)

*Author contributions.* Wolfgang Friederich developed the initial idea of the project. Marcel Paffrath developed the code and ran the calculations. Marcel Paffrath prepared the article with contributions from Wolfgang Friederich.

*Competing interests.* The authors declare that they have no conflict of interest.



*Acknowledgements.* We greatly acknowledge the contributions of the AlpArray temporary network Z3 (Hetényi et al., 2018) and AlpArray Seismic Network (2015) making this work possible. We also want to thank the Deutsche Forschungsgemeinschaft (DFG) for funding the work within the framework of DFG Priority Programme "Mountain Building Processes in Four Dimensions (MB-4D)" (SPP 2017). Many thanks to Tobias Diehl for providing us access to his crust and upper mantle tomography model.

Also, we want to acknowledge all permanent and other temporary seismic networks used in this study: 1N - Malet, J.-P. and Hibert, C. and Radiguet, Mathilde and Gautier, Stéphanie and Larose, Eric and Amitrano, David and Jongmans, Denis and Bièvre, Grégory and RESIF (2015); BW - Department Of Earth And Environmental Sciences, Geophysical Observatory, University Of Munchen (2001); C4 - CERN (2017); CH - Swiss Seismological Service (SED) At ETH Zurich (1983); CR - University Of Zagreb (2001); CZ - Institute Of Geophysics, Academy Of Sciences Of The Czech Republic (1973); FR - RESIF (1995); G - Institut de Physique du Globe de Paris (IPGP) and Ecole et Observatoire des Sciences de La Terre de Strasbourg (EOST) (1982); GE - GEOFON Data Centre (1993); GR - Federal Institute for Geosciences and Natural Resources (BGR) (1976); GU - University Of Genova (1967); HU - Kövesligethy Radó Seismological Observatory (1992); IU - Albuquerque Seismological Laboratory (ASL)/USGS (1988); IV - INGV Seismological Data Centre (1997); MN - MedNet Project Partner Institutions (1988); MT - French Landslide Observatory Seismological Datacenter / RESIF (2006); NI - OGS (Istituto Nazionale Di Oceanografia E Di Geofisica Sperimentale) And University Of Trieste (2002); OE - ZAMG-Zentralanstalt Für Meterologie Und Geodynamik (1987); OX - OGS (Istituto Nazionale Di Oceanografia E Di Geofisica Sperimentale) (2016); PL - Polish Academy of Sciences (PAN) Polskiej Akademii Nauk (1990); RD - RESIF (2018); RF - University Of Trieste (1993); SI - ZAMG - Central Institute for Meteorology and Geodynamics (2006); SK - ESI SAS (Earth Science Institute Of The Slovak Academy Of Sciences) (2004); SL - Slovenian Environment Agency (2001); ST - Geological Survey-Provincia Autonoma Di Trento (1981); SX - Leipzig University (2001); TH - Jena, Friedrich Schiller University (2009); U - Albuquerque Seismological Laboratory (ASL)/USGS (1988); YI - McKee, Kathleen F and Roman, Diana and Fee, David and Ripepe, Maurizio and AIUPPA, Alessandro and Waite, Gregory (2018); YW - Guéguen, P. and Coutant, O. and Langlais, M. and RESIF (2017); ZH - Deschamps, Anne and Beucler, Eric (2013).

The authors would also like to thank all members of the AlpArray Seismic Network team and the AlpArray SWATH-D field team, mentioned in the team list above. As well as the members of the EASI field team: Jaroslava Plomerová, Helena Munzarová, Ludek Vecsey, Petr Jedlicka, Josef Kotek, Irene Bianchi, Maria-Theresia Apoloner, Florian Fuchs, Patrick Ott, Ehsan Qorbani, Katalin Gribovszki, Peter Kolinsky, Peter Jordakiev, Hans Huber, Stefano Solarino, Aladino Govoni, Simone Salimbeni, Lucia Margheriti, Adriano Cavaliere, John Clinton, Roman Racine, Sacha Barman, Robert Tanner, Pascal Graf, Laura Ermert, Anne Obermann, Stefan Hiemer, Meysam Rezaeifar, Edith Korger, Ludwig Auer, Korbinian Sager, György Hetényi, Irene Molinari, Marcus Herrmann, Saulé Zukauskaité, Paula Koelemeijer, Sascha Winterberg. For more information on the team visit www.alparray.ethz.ch.

A special thanks to the authors of Matplotlib (Hunter, 2007), providing a powerful toolkit for scientific data and map visualization with the help of Basemap.

Last but not least we want to thank the seismology group of the Ruhr-Universität Bochum, which helped to improve the quality of this work by numerous discussions and contributions.



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
