# Peer review of "Teleseismic P-waves at the AlpArray seismic network: Wave fronts, absolute traveltimes and traveltime residuals"

_Solid Earth, 2020_

## Short Comment (SC1) · 2 Jan 2021

That is an interesting study.

Regarding steeply incident (e.g. vertical) wavefronts, this study is maybe of interest to compare:

https://pubs.geoscienceworld.org/ssa/srl/article-abstract/89/5/1698/531026/Reflections-from-the-Inner-Core-Recorded-during-a

It uses a reflection from the inner core on an ultra-dense network in the center of AlpArray, and corrects the arrival times for known mantle and crustal structure (fig 7).

[Figure]

The variations would be of interest to compare to your results, and may help with a geological interpretation.

[Figure]

[Figure]

▲ **Figure 7.** Travel-time delays as picked from time-corrected *PKiKP* arrivals. Time corrections include ak135 *PKiKP* travel times and additional corrections for topography, crustal, and mantle structure (e.g., Fig. 5b). (a) Individual delay time at each station; (b) interpolated and low-pass filtered delay-time variation. Dashed black lines separate large-scale geologic/tectonic domains. A, alpine; B, Bohemian massif (Paleozoic orogenesis); C, Carpathian; D, Dinaric domains (Cenozoic orogenesis); Pa, Po, Mo, deep tertiary sedimentary basin systems (Pannonian, Po, and Molasse basins). The color version of this figure is available only in the electronic edition.

**Fig. 1.**

---

## Short Comment (SC2) · 5 Jan 2021

Dear Marcel and co-authors,

Firstly, congratulations for this interesting and very detailed piece of work. I look forward to seeing the results of the inversion of travel time residuals published.

I have 2 suggestions and 1 question on your preprint.

The 2 suggestions deal with the origin of data and the citations of contributing networks:

1- in section 2, you write that "23 ocean bottom seismometers (were) deployed by the LOBSTER project". This is incorrect and I would be grateful if you could correct that

sentence.

The truth is that the OBS component of the AlpArray seismic network was deployed in the framework of a German-French collaboration funded by the LOBSTER project for the German side and by the AlpArray-FR project for the French side. This OBS component of the AASN included 16 instruments from the German DEPAS and GEOMAR pools (A402A, A404A, A405A, A406A, A409A, A412A, A414A, A415A, A417A, A418A, A420A, A421A, A423A, A428A, A430A, A434A) and 8 instruments from the French INSU and Geoazur pools (A401A, A410A, A413A, A416A, A419A, A422A, A425A, A429A). The total number of instruments with data distributed as part of the Z3 network is therefore 24 and not 23.

Please also correct the reference to LOBSTER in the caption of Fig. 10. This sentence should be modified to something like: The ocean-bottom seismometers of the temporary Z3 network in the Ligurian basin are also...

2- your citations and references to seismic networks do not follow the rules of the Federation of Digital Seismic Networks (FDSN) described in http://www.fdsn.org/pdf/V1.0-21Jul2014-DOIFDSN.pdf. Again, I would be grateful, on behalf of all those who acquire and distribute data, if you could correct these citations.

For example, network 1N should be referred to as "Malet et al. (2015)" and the complete citation is "Malet, J.-P., Hibert, C., Radiguet, M., Gautier, S., Larose, E., Amitrano, D., Jongmans, D., Bièvre, G., & RESIF. (2015). French Landslide Observatory – OMIV (Temporary data) (MT-campagne) (RESIF - SISMOB) [Data set]. RESIF - Réseau Sismologique et géodésique Français. https://doi.org/10.15778/RESIF.1N2015" as indicated in http://www.fdsn.org/networks/detail/1N_2015/.

The website http://www.fdsn.org/networks/ indeed provides all the citations for all networks in a number of formats, including BibTeX.

Finally, the question deals with data preprocessing. In section 2.1, you only mention

bandpass filtering in 2 different frequency bands, the high-frequency one being se-lected specifically to reduce oceanic noise on OBS records. Does this mean that you did not correct the records for instrument response? If not, did you check that the re-sponses of the broad variety of instruments used in the dataset are flat in the 0.03-0.1 Hz band?

Thank you in advance for taking these remarks into consideration.

---

## Short Comment (SC3) · 7 Jan 2021

Dear Michael Behm,

thank you very much for your comment. I will have a look at your work to see if there are similarities in the wavefronts. However, since we are looking at different phases with different travel paths, these might be hard to find. Is there a digital version of the traveltime delays you show in Fig. 7 of your work available? This would make a comparison a lot easier for me.

Regards,

[Figure]

Marcel Paffrath

---

## Short Comment (SC4) · 7 Jan 2021

Dear Anne,

thank you very much for your comments and suggestions. I am sorry that I made some mistakes regarding network citations and the correct references to the OBS projects and will correct these in a future iteration!

Regarding your question on data pre-processing: We corrected all data for instrument response before picking. There should have been a sentence in the manuscript stating this. I will correct that as well.

[Figure]

Again, thank you very much for your input.

Regards,

Marcel

---

## Referee Comment (RC1) · Anonymous Referee #1 · 11 Jan 2021

This research work is the data preparation for seismic tomography of the region beneath the AlpArray Seismic network. The authors have presented how the absolute traveltimes and differential traveltimes are obtained. I am quite interested in reading these details. I only have one suggestion. Please include one geologic map to show the tectonic settings of the study area, showing the color-coded seismic networks. The others are some minor suggestions or corrections.

1. In the title it may be better to say differential traveltimes instead of traveltime residuals. Many readers consider traveltime residuals as the difference between the observed and theoretical traveltimes.

[Figure]

2. Line 3. Teleseismic delay time tomography is not often used or seems to be informal. I would prefer teleseismic traveltime tomography.

3. Line 146. Delta tau j B is not defined or explained as well as Delta T j B.

3. Line 183. the signal-to-noise ratio and and the difference. Remove the extra and.

I am looking forward to seeing your tomographic results.

---

## Author Comment (AC1) · 20 Jan 2021

Dear Referee #1,

Thank you very much for your comments and suggestions. We will include a geological map (Fig. 1) showing the tectonic setting and stations. We use the term traveltime residuals, as we are indeed calculating differences of demeaned observed traveltimes to demeaned theoretical ones and not time differences between stations.

We will include the remaining smaller suggestions and corrections as well.

Best regards,

[Figure]

Marcel Paffrath

[Figure]

**Seismic Networks**
- AlpArray
- EASI
- SWATH-D
- OBS
- Permanent (other)

**Tectonic Units**
- Adria accreted
- Adria autochton
- Europe accreted
- Neotethys
- Alpine tethys
- Flexural foredeep and graben fill

**Faults**
- Deformation front, exposed
- Deformation front, subsurface
- Important Neogene fault

**Fig. 1.**

---

## Referee Comment (RC2) · Anonymous Referee #2 · 28 Jan 2021

General Comments:

In their study, Paffrath and co-authors present a strategy for an automated, combined estimation of absolute and relative P-wave arrival times of teleseismic waveforms for the AlpArray seismic network. The authors apply their method to 370 teleseismic earthquakes recorded by the AlpArray network, which includes 1025 permanent and temporary broad-band stations (including OBS stations in the Ligurian sea). The method is applied in two different frequency bands: 1. HF (Band-pass: 0.03-0.5 Hz (2 s – 33 s)), 2. LF (Low-pass: 0.1 Hz (10 s)). The HF dataset contains about 170'000 arrivals, the LF set contains 214'000 arrivals. The authors analyse and interpret the correspond-

ing travel-times and corresponding residuals in terms of quality of the arrival times as well as potential velocity anomalies in the mantle. From the stacked residuals as well as stacks of azimuthal bins, the authors infer the presence and orientation of slabs beneath the Alpine orogen. In addition, the authors analyse and interpret travel-time differences between the two frequency bands and conclude that the observed difference is related to "finite-frequency effects". Travel-times are planned to be used for a future tomographic inversion of the seismic velocity structure of the mantle.

The combined picking-procedure proposed by the authors is reasonable and the derived quality documented by the consistency of the derived travel-time fields and residuals seems very promising. The authors did a great job in attempting to quantify the errors of their arrival times (for the absolute picking as well as the cross-correlation differential time estimation) and their results (and their documented consistency) is quite impressive, hopefully leading to new high-quality images of the mantle structure beneath the Alps. Nevertheless, I have several comments listed below, which hopefully help to improve the quality of this study and presumably require at least moderate revision of the manuscript.

Specific comments:

1) My main concern relates to the interpretation of the presented travel-time residuals. The authors argue that the residuals are mainly related to mantle structure and in the discussion and interpretation residuals are mainly associated with the presence of lithospheric slabs and possible impact of crustal structures are largely ignored/faded out. According to the authors, the argument for this is that residuals calculated for the crustal model of Diehl et al. 2009 (shown in their Figure A2) look different and are of smaller amplitude. The comparison of absolute amplitudes, however, is difficult, since no colour-scale for the residuals is included in Figure A2. In addition, it is not clear to what reference 1D model the residuals shown in Figure A2 relate. Also, what is meant with "near-surface model" (caption in Figure A2)? Is it just the shallowest part of the crustal model of Diehl et al. 2009 or the entire crustal model? Most importantly,

the distribution of teleseismic residuals related to crustal structure (compared to a 1D crust) shown in Figure A2 deviates from other estimates e.g. the one of Waldhauser et al. 2002 (Figure 6 in that paper). Waldhauser et al's model predicts negative residuals <-0.5 s along the western Alpine Arc related to the Ivrea Body and strong positive residuals (>1.0 s) related to the Po-plane sediments. Both signals seem to completely missing in Figure A2 (although at least the Ivrea anomaly is completely included in the crustal model of Diehl et al. 2009). Also, the Molasse sediments in the northern foreland are not visible in A2. It seems to me that the residuals shown in A2 mainly correlate with the crustal thickness (negative where relatively thinner crust, positive where relatively thicker (e.g. crustal root of the Alps/Apennines) but other effects (e.g. sedimentary basins, Ivrea body) are missing. Especially the Ivrea body could have quite a huge impact and I would assume that parts of the negative residuals in the western Alpine Arc (the region labelled "W" e.g. in Fig. 6) are affected by the Ivrea body rather than evidence for a slab. Ivrea is not mention in the entire manuscript although it is expected to be one of the strongest shallow anomalies in the Alps. Some parts in the discussion of the observed residuals also seem inconsistent. E.g. on page 19 line 400 the authors say the "W" anomaly should be shallow (which would be consistent with the Ivrea body), then on page 25 (line 531) they associate it with the subducted European lithosphere (which I would not describe as a shallow anomaly).

Also, I do not fully understand the meaning of the ("demeaned") residuals shown in the different figures and if (and how) they are comparable in absolute terms. The authors write on page 15 (around line 335) that "the stacked residuals are relative to an unknown 1D earth model [...] and not to a standard earth model". Does the comparison of absolute residuals with the ones of figure A2 then makes sense? How different is the "unknown 1D model" from the 1D model assumed for figure A2? If comparable, it would then make more sense to subtract the predicted crustal part (A2) from residuals shown in other figures (e.g. Figure 6 etc). Then the corrected residuals would reflect pure mantle signals (assuming the crustal corrections are correct and complete). Why are the colour bars in most Figures (e.g. 6, 7) labelled as "average

time difference" and not "Travel-time residuals" (as used in the text)?

The authors spent a lot of thoughts and work into the error estimation of their automatic picks. But they do not seem to use this information for the residual analysis and their interpretation. In my opinion it would be more consistent to have section 5 (the description of error distribution) in front of section 4 (the section with the results) and use the observational weights for the stacking of the residuals to get quality-weighted stacks. In the current version, as far as I understand, no weighting is used for the stacked residuals, right?

In summary, the discussion of the observed travel-time residuals should be improved and extended and crustal anomalies possibly affecting the observed residuals (including conclusions of others like Waldhauser et al 2002 or Zhao et al. 2016) should be properly discussed. Also, the authors should address specifically the open questions of Alpine mantle structure and what their preliminary results could potentially contribute to solve them. E.g. Western Alps: Continuous vs detached slab (Lippitsch, Zhao), Eastern Alps: slab reversal (Lippitsch, Mitterbauer). E.g. what is the interpretation of the positive residuals in SE-France? Does this correlate with any feature of previously published models?

On the other hand, since the authors obviously also work on a tomographic inversion of this data set, the whole "preliminary" interpretation of the "raw" residuals could also be drastically shortened and residuals and travel-time fields can be discussed primarily with the goal to document the consistency of their derived travel-time data.

2) The description of the picking procedure is a bit difficult to follow in some places. Maybe the description could be supported by a flow-diagram summarizing the key steps of the proposed method and/or a sketch figure with a generic seismogram illustrating some of the key parameters such as tmpp, tepp, tlpp, etc. Also, I miss some key references in section 3, which indicate that the proposed method is not entirely new but has used/proposed already by others (in slightly different forms). Only in the discussion

the authors mention two papers (Rawlinson & Kennett, Mitterbauer) using a similar approach. But there are other groups using the same idea of CC, stacking, absolute pick on stack and then correcting absolute picks with the CC-information. One of them was proposed by Rowe and co-workers (e.g. Rowe et al 2002, BSSA (An Automatic, Adaptive Algorithm for Refining Phase Picks in Large Seismic Data Sets). Also "classic" references to AIC-pickers, Higher-Order statistics and beamforming should be added when introduced the first time in the manuscript. In the discussion, the authors should also compare the performance of their processing in a more quantitative matter with other studies. E.g. how absolute and relative uncertainties compare with other teleseismic data sets in the Alps (Lippitsch, Mitterbauer, Zhao) or elsewhere. Additionally, useful information to be discussed: How computationally expensive is their method? How long does it take to process this dataset? Is the code to parallelized? What are the computational requirements? Is the code published and usable for others?

3) I understand that the difference in travel-time residuals between HF and LF data is interpreted as "finite-frequency" effect (e.g. page 26, l. 565) by the authors. In the abstract, however, the authors write, "caused by velocity dispersion". Somewhere else (section 4.4) the authors explain "velocity dispersion due to attenuation" and then mention "finite frequency effect (finite wavelength effect)" as alternative explanation, which seems to relate to the different physical resolution due difference in frequency content. Later in that section the authors mention the effect of "wavelength smoothing". None of these mentioned "effects" comes with a reference and I find it confusing what "effect" the authors finally prefer to explain the observed difference in travel times. I interpret "finite-frequency effect" as the "wavefront-healing effect" e.g. described by Wielandt 1987 or Hung et al 2001. Is this what the authors mean? I would suggest to use a consistent and homogeneous terminology in all parts of the manuscript and to improve the description of potential effects (including references describing these effects).

4) The English should be further improved in my opinion. I spotted several issues in
grammar, punctuation, and style, some examples are listed below. In general, I found superfluous repetitions as well as vague and rather qualitative statements (e.g. "highly important", "highly accurate", "drastically higher signal-to-noise ratio") in many places of the manuscript, which should be improved in another round of proofreading. Try to replace these qualitative statements with quantitative ones if possible (e.g. "results in improvement of signal-to-noise by a factor of X")

Detail comments:

- l. 5: At this point it is not clear what the frequencies relate to (high-pass or low-pass, lower or upper corner?). - l. 10: "reproducible" -> "stable"?

- l. 17: ". . . way lower" -> "X times smaller" or " a factor of X smaller"

- l. 19: "location dependant" what does that mean? Site-specific noise at station or depending on the source (earthquake) location (e.g. shallow versus deep)?

- l. 25: "At its core" -> "The backbone"?

- l. 27: -> "river Main in Germany"

- l. 35: -> add references for tele-tomo methods

- l. 36: -> explain why resolution is limited to 500 km depth (network aperture controls cross-firing at depth) or give reference

- l. 40: I would mainly argue that FWI needs a good (3-D) starting model and therefore requires knowledge from travel-time tomo

- l. 50: "oscillatory" is there a better word? E.g. "monochromatic"? "emergent"?

- l. 64: "earthquake location" -> I would say that these picking methods work well on lo-cal to regional scales and have been used for earthquake location and local earthquake tomography methods.

- l. 65: it sounds like tele-tomo requires higher precision than local and regional

methods... I would rather argue that CC can be used for tele-tomo because of the homogeneous simple waveforms from a close to planar wave-front incident from below. Therefore, CC is not usable for local, regional methods, unless you have similar earthquakes (e.g. used in relative relocation).

- l. 75: "Using AlpArray data, we demonstrate that..."

- l. 81: "... prior to any tomographic inversion ... in the upper mantle."

- l. 85: "... network Z3 started in 2015." Also: somewhere here I would expect the reference to a Figure showing this network, could be combine with existing Figure 1 (a=network map, b=event distribution)

- l. 92: "Moho jump" -> "Moho offset" Also: Add references for all these possible structures in Moho and slab geometry.

- l. 96: -> "peak in station..."

- l. 102: You used the gCMT catalogue, did you use the centroid-time as origin time? This time might be different to the origin-time derived from P-waves (e.g. used by NEIC or ISC), right? Could that explain some of the larger differences between predicted and observed P-onsets you report later? Why not use a body-wave based OT as reported by NEIC or ISC catalogues?

- l. 105: What order did you use for your filters? Did you make sure that phase-shifts are avoided? Zero-phase filter? Can you exclude any impact of filters to the difference in the two data sets?

- l. 118: what do you mean with these "statistical anomalies"? Anomalies in terms of what?

- l. 140: "low-noise beam trace" add a reference in which beamforming is described

- l. 160: "higher-order ...and the Akaike" add references where this has been described and used for picking.

- l. 165: somewhere it has to be clearly mentioned that this HOS/AIC picker is used for the individual traces (to get a time reference for the CC) as well as for the absolute pick derived from the stacked seismogram. At least this is how I understand it. . . What is the advantage of applying AIC to the HOS-CF rather than directly on the seismogram? Shouldn't the AIC work as well on the original seismogram? Using the HOS-CF might introduce systematic shifts due to the finite window length used for the moving-window approach in the HOS-CF calculation. Can you exclude delays caused by window length? Is the window-length fixed or dynamic (considering the frequency content of the signal)?

- l. 198: You mentioned "manually evaluated". Did you compare picks and uncertainties to a subset of manually determined onsets at some point to assess the quality and reliability of this approach?

- l. 199: "The reason for this is . . ."

- l. 205 and elsewhere: "anchor point" -> "time reference" or "reference time"

- l. 210: Again, is it possible that the large time offset between predicted and actual onset is due to the fact that you use the gCMT centroid OT? Would that be the same if you use NEIC or ISC times?

- l. 222: "foots on" -> "is based on the assumption"

- l. 234: "as representer" -> "to be representative of"

- l 244: not clear, did you pick the onset on the beam by hand or automatically (or both?) in your study? If automatic, it's the algorithm described in 3.2, correct?

- Figure caption 3: When reading the caption, it is not yet clear that there are uncertainties for the absolute as well as relative onsets calculated (the description of the relative only comes later). Maybe say "Onset uncertainties for absolute and relative onsets (as described in the text) are displayed by . . ."

- l 247: It's the first time the traces are correlated with the beam, not the second time, right? "The traveltime of. . ." -> "the traveltime at. . ."

- l. 252: delete "by construction"

- l. 253 and elswhere: "jitter" -> "scatter"

- l. 271: "any signs" -> "any evidence"

- l 285: Does your error assessment for CC-delays also identifies possible problems with cycle-skipping? This is particular problematic for emergent onsets. A potential sign for this is neighbouring local peaks in the CC-CF. Is your algorithms able to identify this?

- l. 286: not sure "significant coda" is the right term here. It is more additional complexity in the signal, e.g. caused by converted phases. It's not really a long lasting "coda". . .

- Figure 3: Should be bigger (portrait arrangement), in caption say that this is one event (and which one). "heavy outlier" -> "severe"? how is this outlier defined? Here and elsewhere in text and other captions: "temporal distance of 1 s": This sound quite wrong to me. You should rephrase this, e.g. "isochrone contour intervals of 1 s" or something. . .

- l. 293: "superiority" -> please rephrase that e.g. "demonstrates the improvement"

- l. 297: shouldn't it be NE to SW?

- Figure 5: Put azimuth and distance as bold text in the figure itself, this makes it easier to compare (otherwise one needs to find this in the caption).

- l. 310: isn't it a obvious fact that the larger the epicentral distance the steeper the incidence, the higher the apparent velocity?

- l. 314 and elsewhere: "widening" -> "broadening" ?

- l. 335: "find out" -> "identify"

- l. 335: "It is highly important. . ." Please rephrase that sentence, it's also not clear to me what you mean here (see my general comment on comparison of residuals).

- l. 334: "mantle events" -> you mean "mantle phases", right? Not earthquakes in the mantle. . .

- l. 346: "completely overwhelm" -> rephrase, maybe "dominate over data from poorly. . ." - Figure 6: Make the inset bigger (currently it has the same size as the original data, not clear what's the benefit of it). What is "structures Vp values. . ."? Where is the tectonic map of M. Handy you mention in the caption?

- l. 378: "Features that start to move" please rephrase. . . the are "laterally shifted" or something.

- l. 382: "reinforced" -> "enhanced"?

- l. 385 "OBS" -> "OBS stations"

- l. 395: "survive" -> rephrase, maybe "remain"?

- Figure 9: Too small, add color bar. . .

- l. 484: "positive impact . . . is salient" Not sure, I don't see this in Figure 10, symbols are too small, make Figure 10 bigger. . .

- l. 496: "have tightened" -> please rephrase

- l. 497: "A way out of this dilema" -> please rephrase. . . "To overcome this problem. . ."

- l. 532: "We do not carry out here. . ." -> please rephrase

- Discussion: Please carefully revise the English in this part. . .

---

## Author Comment (AC2) · 23 Feb 2021

*General Comments:*

*In their study, Paffrath and co-authors present a strategy for an automated, combined estimation of absolute and relative P-wave arrival times of teleseismic waveforms for the AlpArray seismic network. The authors apply their method to 370 teleseismic earthquakes recorded by the AlpArray network, which includes 1025 permanent and temporary broad-band stations (including OBS stations in the Ligurian sea). The method is applied in two different frequency bands: 1. HF (Band-pass: 0.03-0.5 Hz (2 s – 33 s)), 2. LF (Low-pass: 0.1 Hz (10 s)). The HF dataset contains about 170'000 arrivals, the LF set contains 214'000 arrivals. The authors analyse and interpret the corresponding travel-times and corresponding residuals in terms of quality of the arrival times as well as potential velocity anomalies in the mantle. From the stacked residuals as well as stacks of azimuthal bins, the authors infer the presence and orientation of slabs beneath the Alpine orogen. In addition, the authors analyse and interpret travel-time differences between the two frequency bands and conclude that the observed difference is related to "finite-frequency effects". Travel-times are planned to be used for a future tomographic inversion of the seismic velocity structure of the mantle.*

*The combined picking-procedure proposed by the authors is reasonable and the derived quality documented by the consistency of the derived travel-time fields and residuals seems very promising. The authors did a great job in attempting to quantify the errors of their arrival times (for the absolute picking as well as the cross-correlation differential time estimation) and their results (and their documented consistency) is quite impressive, hopefully leading to new high-quality images of the mantle structure beneath the Alps. Nevertheless, I have several comments listed below, which hope-fully help to improve the quality of this study and presumably require at least moderate revision of the manuscript.*

Thank you very much for your comments, suggestions and the very detailed discussion of our work.

*Specific comments:*

*1) My main concern relates to the interpretation of the presented travel-time residuals. The authors argue that the residuals are mainly related to mantle structure and in the discussion and interpretation residuals are mainly associated with the presence of lithospheric slabs and possible impact of crustal structures are largely ignored/faded out. According to the authors, the argument for this is that residuals calculated for the crustal model of Diehl et al. 2009 (shown in their Figure A2) look different and are of smaller amplitude. The comparison of absolute amplitudes, however, is difficult, since no colour-scale for the residuals is included in Figure A2. In addition, it is not clear to what reference 1D model the residuals shown in Figure A2 relate. Also, what is meant with "near-surface model" (caption in Figure A2)? Is it just the shallowest part of the crustal model of Diehl et al. 2009 or the entire crustal model? Most importantly, the distribution of teleseismic residuals related to crustal structure (compared to a 1D crust) shown in Figure A2 deviates from other estimates e.g. the one of Waldhauser et al. 2002 (Figure 6 in that paper). Waldhauser et al's model predicts negative residuals <-0.5 s along the western Alpine Arc related to the Ivrea Body and strong positive residuals (>1.0 s) related to the Po-plane sediments. Both signals seem to be completely missing in Figure A2 (although at least the Ivrea anomaly is completely included in the crustal model of Diehl et al. 2009). Also, the Molasse sediments in the northern foreland are not visible in A2. It seems to me that the residuals shown in A2 mainly correlate with the crustal thickness (negative where relatively thinner*

*crust, positive where relatively thicker (e.g. crustal root of the Alps/Apennines) but other effects (e.g. sedimentary basins, Ivrea body) are missing. Especially the Ivrea body could have quite a huge impact and I would assume that parts of the negative residuals in the western Alpine Arc (the region labelled "W" e.g. in Fig. 6) are affected by the Ivrea body rather than evidence for a slab. Ivrea is not mention in the entire manuscript although it is expected to be one of the strongest shallow anomalies in the Alps. Some parts in the discussion of the observed residuals also seem inconsistent. E.g. on page 19 line 400 the authors say the "W" anomaly should be shallow (which would be consistent with the Ivrea body), then on page 25 (line 531) they associate it with the subducted European lithosphere (which I would not describe as a shallow anomaly).*

We did not plan to interpret the traveltime residuals in detail in the first place and mainly wanted to show the data and residual patterns as a documentation of our work prior to a traveltime tomography, without making assumptions on e.g. crustal structures, knowing that the model by Diehl is not complete, especially in areas with low station and/or event coverage (e.g. in the Po-plain) and does not extend over the whole study area of AlpArray. Therefore, we mainly wanted to show that the residual pattern we get by calculating residuals through the regional tomography model by T. Diehl has a very different shape and the residuals are mainly influenced by mantle structures. Upon request of the editor, we extended the interpretation of our results and tried to correlate them with possible anomalies, mainly in the upper mantle.

However, we are now aware that the effect of the crustal anomalies should be discussed in more detail when trying to interpret residuals as mantle-dominated as the crust of course has a great impact on the overall residuals and the calculation of crustal residuals of the Diehl model is not sufficient. Due to our simultaneous work on the tomography, we meanwhile created a more complete crustal model using additional information from the EuCRUST-07 model (Tesauro et al. 2008) and Moho information of Spada et al. (2013). The new model also includes sedimentary basins such as the Po-plain and the Molasse basin. Hence, we decided to correct the stacked residuals we show using information of this new model (assuming a horizontal planar wavefront, only crustal part of Diehl) and also present a residual map of this model relative to the minimum 1D model of Diehl et al., comparable to Fig. 6 of Waldhauser et al. (2002). As a consequence, we also modified our interpretations of the resulting maps of traveltime residuals.

*Also, I do not fully understand the meaning of the ("demeaned") residuals shown in the different figures and if (and how) they are comparable in absolute terms. The authors write on page 15 (around line 335) that "the stacked residuals are relative to an unknown 1D earth model [. . .] and not to a standard earth model". Does the comparison of absolute residuals with the ones of figure A2 then makes sense? How different is the "unknown 1D model" from the 1D model assumed for figure A2? If comparable, it would then make more sense to subtract the predicted crustal part (A2) from residuals shown in other figures (e.g. Figure 6 etc). Then the corrected residuals would reflect pure mantle signals (assuming the crustal corrections are correct and complete). Why are the colour bars in most Figures (e.g. 6, 7) labelled as "average time difference" and not "Travel-time residuals" (as used in the text)?*

In principle, the true 1D model is different for each event. However, since for teleseismic waves incidence angles are rather steep waves from all events see most of the mantle and crust beneath the Alps before reaching the stations. Thus, we may hope that the true 1D models only differ marginally from each other. If this is true then the residuals represent structural perturbations from this 1D model which average to zero. We think it makes sense to subtract residuals for a crustal model if the 1D model assumed for the crust is the average of the crustal 3D model and the residuals average to zero over the array.

We changed the colour bar labels to "traveltime residuals".

*The authors spent a lot of thoughts and work into the error estimation of their automatic picks. But they do not seem to use this information for the residual analysis and their interpretation. In my*

*opinion it would be more consistent to have section 5 (the description of error distribution) in front of section 4 (the section with the results) and use the observational weights for the stacking of the residuals to get quality-weighted stacks. In the current version, as far as I understand, no weighting is used for the stacked residuals, right?*

We interchanged section 4 and 5 and also followed your suggestion of weighting the residuals in the stacking process by the pick uncertainty.

*In summary, the discussion of the observed travel-time residuals should be improved and extended and crustal anomalies possibly affecting the observed residuals (including conclusions of others like Waldhauser et al 2002 or Zhao et al. 2016) should be properly discussed. Also, the authors should address specifically the open questions of Alpine mantle structure and what their preliminary results could potentially contribute to solve them. E.g. Western Alps: Continuous vs detached slab (Lippitsch, Zhao), Eastern Alps: slab reversal (Lippitsch, Mitterbauer). E.g. what is the interpretation of the positive residuals in SE-France? Does this correlate with any feature of previously published models?*

*On the other hand, since the authors obviously also work on a tomographic inversion of this data set, the whole "preliminary" interpretation of the "raw" residuals could also be drastically shortened and residuals and travel-time fields can be discussed primarily with the goal to document the consistency of their derived travel-time data.*

We slightly modified the structural interpretation of the traveltime residuals trying to find a compromise between your suggestions and those of the topical editor. We want to keep the interpretation because we find it interesting and also surprising how many conclusions can be drawn from the "raw" residuals only (in particular from the shifting of the patterns with varying azimuth). They provide a useful guideline when judging the plausibility of a tomographic model derived from them.

*2) The description of the picking procedure is a bit difficult to follow in some places. Maybe the description could be supported by a flow-diagram summarizing the key steps of the proposed method and/or a sketch figure with a generic seismogram illustrating some of the key parameters such as tmpp, tepp, tlpp, etc. Also, I miss some key references in section 3, which indicate that the proposed method is not entirely new but has used/proposed already by others (in slightly different forms). Only in the discussion the authors mention two papers (Rawlinson & Kennett, Mitterbauer) using a similar approach. But there are other groups using the same idea of CC, stacking, absolute pick on stack and then correcting absolute picks with the CC-information. One of them was proposed by Rowe and co-workers (e.g. Rowe et al 2002, BSSA (An Automatic, Adaptive Algorithm for Refining Phase Picks in Large Seismic Data Sets). Also "classic" references to AIC-pickers, Higher-Order statistics and beamforming should be added when introduced the first time in the manuscript. In the discussion, the authors should also compare the performance of their processing in a more quantitative matter with other studies. E.g. how absolute and relative uncertainties compare with other teleseismic data sets in the Alps (Lippitsch, Mitterbauer, Zhao) or elsewhere. Additionally, useful information to be discussed: How computationally expensive is their method? How long does it take to process this dataset? Is the code to parallelized? What are the computational requirements? Is the code published and usable for others?*

We have added a flow diagram of the picking algorithm in the revised manuscript, showing the different steps of the algorithm and also included further references.

We also included a qualitative comparison of the uncertainties of our data with those of other data sets used in previous tomographic work on Alpine mantle structure. However, not all of these publications give detailed information on their pick uncertainties, neither in a quantitative nor in a qualitative way (i.e. how they are estimated).

The code is parallelized, but not completely optimized yet. It was running on a ~5 year old 20 CPU machine, where the computation time of one correlation onset including error estimation for a single station averages at about 0.5s. For the full $db_{0.5}$ dataset computation was finished in less than a day.

*3) I understand that the difference in travel-time residuals between HF and LF data is interpreted as "finite-frequency" effect (e.g. page 26, l. 565) by the authors. In the abstract, however, the authors write, "caused by velocity dispersion". Somewhere else (section 4.4) the authors explain "velocity dispersion due to attenuation" and then mention "finite frequency effect (finite wavelength effect)" as alternative explanation, which seems to relate to the different physical resolution due difference in frequency content. Later in that section the authors mention the effect of "wavelength smoothing". None of these mentioned "effects" comes with a reference and I find it confusing what "effect" the authors finally prefer to explain the observed difference in travel times. I interpret "finite-frequency" effect as the "wavefront-healing effect" e.g. described by Wielandt 1987 or Hung et al 2001. Is this what the authors mean? I would suggest to use a consistent and homogeneous terminology in all parts of the manuscript and to improve the description of potential effects (including references describing these effects).*

We reworked the part dealing with the observed frequency dependence. As "finite-frequency effect" we refer to wave propagation phenomena that deviate from the predictions of ray theory which is a zero-wavelength approximation. One example is the wavefront healing effect described by Wielandt but also diffractions or interference phenomena are an expression of the finite wavelength of real seismic waves. Dispersion by attenuation refers to the fact that wave velocities become frequency dependent in attenuating media. It is different from geometric dispersion created when waves propagate in finite bodies (e. g. surface wave dispersion). In the revised manuscript, we now use the terms "finite-frequency effect" and "dispersion by attenuation" only. Moreover, we have slightly reduced the discussion about dispersion by attenuation because the observed average bias between high and low-frequency travel times can most easily be explained by the group delay of the applied filter. On the other hand, we show that the (demeaned) residuals are not affected by the filter.

*4) The English should be further improved in my opinion. I spotted several issues in grammar, punctuation, and style, some examples are listed below. In general, I found superfluous repetitions as well as vague and rather qualitative statements (e.g. "highly important", "highly accurate", "drastically higher signal-to-noise ratio") in many places of the manuscript, which should be improved in another round of proofreading. Try to replace these qualitative statements with quantitative ones if possible (e.g. "results in improvement of signal-to-noise by a factor of X")*

We went through the text again in detail and tried to improve the mentioned statements. There should be a proofreading by the publisher in the typesetting process.

*Detail comments:*

*-        l. 5: At this point it is not clear what the frequencies relate to (high-pass or low-pass, lower or upper corner?). - l. 10: "reproducible" -> "stable"?*

Done

*-        l. 17: ". . . way lower" -> "X times smaller" or "a factor of X smaller"*

Estimated the SNR factor

*-        l. 19: "location dependant" what does that mean? Site-specific noise at station or depending on the source (earthquake) location (e.g. shallow versus deep)?*

Site-specific noise!

- l. 25: "At its core" -> "The backbone"?

Done

- l. 27: -> "river Main in Germany"

Done

- l. 35: -> add references for tele-tomo methods

Done

- l. 36: -> explain why resolution is limited to 500 km depth (network aperture controls cross-firing at depth) or give reference

Done

- l. 40: I would mainly argue that FWI needs a good (3-D) starting model and therefore requires knowledge from travel-time tomo

Absolutely! It got lost somewhere in the manuscript preparation process.

- l. 50: "oscillatory" is there a better word? E.g. "monochromatic"? "emergent"?

monochromatic

- l. 64: "earthquake location" -> I would say that these picking methods work well on local to regional scales and have been used for earthquake location and local earthquake tomography methods.

Done

- l. 65: it sounds like tele-tomo requires higher precision than local and regional methods... I would rather argue that CC can be used for tele-tomo because of the homogeneous simple waveforms from a close to planar wave-front incident from below. Therefore, CC is not usable for local, regional methods, unless you have similar earthquakes (e.g. used in relative relocation).

That was not our intention and we changed this part.

- l. 75: "Using AlpArray data, we demonstrate that. . ."

Done

- l. 81: ". . . prior to any tomographic inversion ... in the upper mantle."

Done

- l. 85: ". . . network Z3 started in 2015." Also: somewhere here I would expect the reference to a Figure showing this network, could be combine with existing Figure 1 (a=network map, b=event distribution)

Done, added figure reference

- l. 92: "Moho jump" -> "Moho offset" Also: Add references for all these possible structures in Moho and slab geometry.

Done

- l. 96: -> "peak in station. . ."

Done

- l. 102: You used the gCMT catalogue, did you use the centroid-time as origin time? This time might be different to the origin-time derived from P-waves (e.g. used by NEIC or ISC), right? Could that explain some of the larger differences between predicted and observed P-onsets you report later? Why not use a body-wave based OT as reported by NEIC or ISC catalogues?

Yes we used centroid times as onset times in our database, as we also want to use them for a later FWI. Differences to the theoretical onsets we calculate can also be influenced by the difference between the origin times. We made this clear in the text now.

- l. 105: What order did you use for your filters? Did you make sure that phase-shifts are avoided? Zero-phase filter? Can you exclude any impact of filters to the difference in the two data sets?

We use $4^{th}$ order butterworth bandpass filtering. Hence, there will be time shifts that are frequency dependent. They can indeed explain easily the observed average bias between the traveltimes picked at different frequencies. But we show in the revised version that the (demeaned) traveltime residuals are not affected by the filter because the group delay cancels when subtracting the array average.

- l. 118: what do you mean with these "statistical anomalies"? Anomalies in terms of what?

We meant features that are conspicuous. However, we removed this from the manuscript.

- l. 140: "low-noise beam trace" add a reference in which beamforming is described

Done

- l. 160: "higher-order . . .and the Akaike" add references where this has been described and used for picking.

Done

- l. 165: somewhere it has to be clearly mentioned that this HOS/AIC picker is used for the individual traces (to get a time reference for the CC) as well as for the absolute pick derived from the stacked seismogram. At least this is how I understand it. . . What is the advantage of applying AIC to the HOS-CF rather than directly on the seismogram? Shouldn't the AIC work as well on the original seismogram? Using the HOS-CF might introduce systematic shifts due to the finite window length used for the moving-window approach in the HOS-CF calculation. Can you exclude delays caused by window length? Is the window-length fixed or dynamic (considering the frequency content of the signal)?

We would kindly refer you to the paper by Kueperkoch et al. (2010) for those detailed questions, as we are using their algorithm for the AIC onsets.

- l. 198: You mentioned "manually evaluated". Did you compare picks and uncertainties to a subset of manually determined onsets at some point to assess the quality and reliability of this approach?

No, we manually validated the picks (by visual inspection) to find out about systematic problems.

- l. 199: "The reason for this is . . ."

Done

- l. 205 and elsewhere: "anchor point" -> "time reference" or "reference time"

Done

- l. 210: Again, is it possible that the large time offset between predicted and actual onset is due to the fact that you use the gCMT centroid OT? Would that be the same if you use NEIC or ISC times?

Yes, included this.

- l. 222: "foots on" -> "is based on the assumption"

Done

- l. 234: "as representer" -> "to be representative of"

Done

- l 244: not clear, did you pick the onset on the beam by hand or automatically (or both?) in your study? If automatic, it's the algorithm described in 3.2, correct?

We used the automatic picking algorithm, added this.

- Figure caption 3: When reading the caption, it is not yet clear that there are uncertainties for the absolute as well as relative onsets calculated (the description of the relative only comes later). Maybe say "Onset uncertainties for absolute and relative onsets (as described in the text) are displayed by . . ."

Made this clearer in the figure caption.

- l 247: It's the first time the traces are correlated with the beam, not the second time, right? "The traveltime of. . ." -> "the traveltime at. . ."

Each station is correlated for the first time with the reference station. After beamforming another correlation is done with the beam trace.

- l. 252: delete "by construction"

Done

- l. 253 and elswhere: "jitter" -> "scatter"

Done

- l. 271: "any signs" -> "any evidence"

Done

- l 285:  Does your error assessment for CC-delays also identifies possible problems with cycle-skipping? This is particular problematic for emergent onsets. A potential sign for this is neighbouring local peaks in the CC-CF. Is your algorithms able to identify this?

The algorithm searches for severe outliers. However, we also visually inspected the traveltime patterns of each event for remaining outliers (there were only a hand full). Also added this in the text.

- l. 286: not sure "significant coda" is the right term here. It is more additional complexity in the signal, e.g. caused by converted phases. It's not really a long lasting "coda". . .

Done

- Figure 3: Should be bigger (portrait arrangement), in caption say that this is one event (and which one). "heavy outlier" -> "severe"? how is this outlier defined? Here and elsewhere in text and other captions: "temporal distance of 1 s": This sound quite wrong to me. You should rephrase this, e.g. "isochrone contour intervals of 1 s" or something. . .

Made this clearer in the text. Also increased figure size and put them below each other.

- l. 293: "superiority" -> please rephrase that e.g. "demonstrates the improvement"

Done

- *l. 297: shouldn't it be NE to SW?*

Yes indeed!

- *Figure 5: Put azimuth and distance as bold text in the figure itself, this makes it easier to compare (otherwise one needs to find this in the caption).*

Done

- *l. 310: isn't it a obvious fact that the larger the epicentral distance the steeper the incidence, the higher the apparent velocity?*

Yes this is true. However, we wanted to highlight this fact for a broader audience with non-seismological background as well.

- *l. 314 and elsewhere: "widening" -> "broadening"?*

Done

- *l. 335: "find out" -> "identify"*

Done

- *l. 335: "It is highly important. . ." Please rephrase that sentence, it's also not clear to me what you mean here (see my general comment on comparison of residuals).*

See above

- *l. 334: "mantle events" -> you mean "mantle phases", right? Not earthquakes in the mantle...*

Yes we do!

- *l. 346: "completely overwhelm" -> rephrase, maybe "dominate over data from poorly. . ." - Figure 6: Make the inset bigger (currently it has the same size as the original data, not clear what's the benefit of it). What is "structures Vp values. . ."? Where is the tectonic map of M. Handy you mention in the caption?*

Rephrased the sentence and increased size of inset. We decreased the point size of the residuals in the inset so that each station of the dense SWATH-D network can be seen individually, as they overlap in the main figure. The tectonic map is in the background of the residuals, but will now be shown separately together with the different networks.

- *l. 378: "Features that start to move" please rephrase. . . the are "laterally shifted" or something.*

Done

- *l. 382: "reinforced" -> "enhanced"?*

Done

- *l. 385 "OBS" -> "OBS stations"*

Done

- *l. 395: "survive" -> rephrase, maybe "remain"?*

Done

- *Figure 9: Too small, add color bar. . .*

Increased Figure size. However, we decided not to add a color bar here which would be redundant as the precise uncertainties are shown in the x-axis of the plot. We changed the caption to clarify that we coloured the histogram for an easier comparison to the following map plot.

- *l. 484: "positive impact . . . is salient" Not sure, I don't see this in Figure 10, symbols are too small, make Figure 10 bigger. . .*

Done

- *l. 496: "have tightened" -> please rephrase*

Done

- *l. 497: "A way out of this dilema" -> please rephrase. . . "To overcome this problem. . ."*

Done

- *l. 532: "We do not carry out here. . ." -> please rephrase*

Done

- *Discussion: Please carefully revise the English in this part. . .*

Done

---

## Author Response (AR2)

General Comments:

In the revised version of their manuscript, Paffrath and co-authors considered all my previously raised concerns. The calculation of crustal corrections and the presentation of the travel-time residuals have been greatly improved. The new Figure A2 suggests that the updated/improved model used for crustal corrections seems much more complete and realistic compared to the one presented in the previous version of the manuscript. I am still in doubts, however, if the current quality of text and English is sufficient for immediate publication in Solid Earth. There are still many flaws in style and punctuation, a lot of repetitions and partly superfluous statements and conjunctives. The manuscript would also benefit from improvements in terms of structure. For instance, the discussion (about 3 pages) is mainly an extensive conclusion and, on the other hand, the results presented in section 5 (>9 pages) already include interpretations I would have expected in the discussion (e.g. section 5.4). I have provided a list with selected examples (and suggestions how to improve) as well as additional minor comments below.

We have modified the structure of the manuscript and moved interpretations in section 5.3 and 5.4 to the discussion (section 6). The text in the discussion referring to the interpretation of the traveltime residuals has been reworked.  Other remarks concerning style and phrasing have been taken into account.

One final comment regarding the interpretation of the residuals: In section 5 the authors provide long and detailed descriptions of observations (L, A, W, C, E in Fig. 10), but the interpretation of these observations (especially in section 6) is extremely short and vague. The main conclusion seems that the expected 3-D mantle structure is "complex" (which is not a surprise based on previous studies). Anomalies W and A are associated with known slabs, but the interpretation of C and E is extremely vague and short in section 6 (see previous comment on structure). Also, I have to admit that I do not fully understand how the authors defined the outlines of anomalies W, C, E in Figure 10 and elsewhere. Especially for C and E it is not clear to me at all. On the other hand, the delays (positive residuals) in the western Alps are not discussed/mentioned at all in section 6, although potentially important in terms of the discussion about slab break-off in this part. I completely understand that this will be part of the author's follow-up tomography study, but I am wondering if all the details in section 5 are really needed for the rather vague interpretation of the residuals.

We note here that the descriptions of the anomalies L, A, W, C, E in the text is just 8 lines long. In addition, definition of these anomalies proved very useful to study the dependency of the location of residual patterns on the azimuth range of the incident waves. The section describing this azimuthal dependence has been shortened and interpretative parts moved to the discussion. The interpretation is as specific as it can be given the fact that only integrative traveltime residuals are available. It still makes remarkable qualitative statements regarding the location, depth and dip of positive velocity perturbations. The definition of the anomalies is to some extent subjective but not arbitrary as the defined areas group the negative residuals and are separated from their neighbours by (narrow) regions of positive or close-to-zero residuals. This is also the case for anomalies C and E as demonstrated in the inset of Fig. 10. That the definition of C and E makes sense is later confirmed by the fact that they show a very different dependence on wave azimuth.  We also have added an interpretative sentence in the discussion regarding the positive residuals in SE France. We do not think that the residuals alone allow statements regarding detachment and slab breakoff because small-scale gaps within regions of positive velocity perturbations will probably not show up as positive traveltime residuals owing to the integrative behaviour of the residuals.

Detail comments:

- l. 12: "that already indicate" -> "that indicate"

Done

- l. 35: "… below the array." add a reference, e.g. Aki et al 1977

Done

- l. 36: "Imagining … " -> "Assuming" or "Considering"?

Done

- l. 40: Why not add a reference in this paragraph, others described the trade-off between array aperture und resolved depth range before, e.g. Sandoval et al 2003 & 2004 (GJI), etc.

Done

- l. 61: "forbidding" -> "unfeasible"?

Done

- l. 73: -> Rowe et al doesn't really relate to teleseismic: -> "In case of similar waveforms, e.g. from earthquake clusters or planar waves of teleseismic wave fronts, one can improve traveltime measurements by … "

Done

- l. 89: -> "… that already indicate the approximate location of high and low velocity anomalies in the upper mantle prior to any tomographic inversion"

Done

- l. 99: -> ground motions

Done

- l. 115: -> "Because oceanic microseismic noise … was only possible for strong earthquakes of magnitudes >XX."

Done, quantified minimum magnitudes (which are of course also distance dependent)

- l. 131: "… capability of algorithms applied to different characteristic functions to resolve … required for high-resolution traveltime tomography. We will summarize the most… "

Done

- l. 134: "characteristic function picking algorithms" -> Rephrase! See above.

Done

- l. 148: "The crucial question is, how to obtain" -> Rephrase!

Done

- l. 151: "read" -> "determine"

Done

- l. 162: "…propagate into" -> "does not affect"

Done

- l. 172: In this paragraph you should add the reference to Figure 3 to give an overview on the procedure before starting with the detailed description of the individual steps.

Done

- l. 178: -> "… which was originally designed…."

Done

- l. 188: what is the "calculation window"?

We rearranged this part to clarify what we mean with calculation window (cut window in which the characteristic functions are calculated)

- l. 191: We select the moving time window…" Meaning of this sentence is still not clear to me… Try to rephrase?

Removed this part as it is not vital for understanding.

- l. 208: -> "Fig 5a…" The order is wrong! Figure 3 and 4 are not mentioned in the text yet. Therfore Figure 5 should be Figure 3 otherwise mention Figure 3 and 4 before.

Add information that Figure 5 will appear later in the text.

- l. 208: The last sentence of this paragraph doesn't make much sense to me. You mean "To resolve the fine-scale mantle structure below the Alps, it is crucial to reduce the uncertainties of the onsets using additional constraints provided by the high station density of the AlpArray network"?

Improved this sentence

- l. 210: -> "By visual inspection of selected examples, we validated that the large uncertainties…"

Done

- l. 212: "is hidden in the site-specific noise"?

Done

- l. 213: "Another limitation…"

Done

- l. 217: -> "Although too uncertain to be used for tomographic inversion, the AIC onsets turned out to be more precise than onsets predicted with standard 1-D earth models".

Done

- l. 233: Meaning of last sentence of paragraph not clear. "Yield" in terms of what? More picks?

Yes, clarified this.

- l. 244: -> "… Alps, within the shortest possible distance to all other stations to minimize …"?

Done

- l. 253: -> "... is ≥0.8 (Fig. 3)." Delete the following sentence.

Done

- l. 257 -> "... can be determined precisely either automatically or manually. In our case, we applied the automatic picking procedure of section 3.2 to determine the onset on the beam trace."

Done

- l. 264 -> "shown in Figure 8" Wrong order as before... Figures 6 and 7 have not been mentioned yet...

Removed Figure reference.

- l. 271: -> "Figure 5c demonstrates... makes them insufficient"

Done

- l. 318: -> "... only about 10% of the total number..."

Done

- l. 320: It is not explained what you are doing with class-4 picks. Since they don't have an upper error-bound (only a lower bound) these picks should be rejected for tomography. This should be mentioned.

Explained this in the paper

- Figure caption 7: "are identifiable" -> "are characterized by"

Done

- l 331: "... in one dataset are not present..."

Done

- l. 377: -> "A closer examination of traveltime residuals shown in Figure 9b and 9d reveals that..."

Done

- l. 400: "influence of incident angle is small" -> but doesn't Figure 9b and 9c show otherwise?

We changed this part to explain what we mean here. There is certainly an influence of the distance, but its influence on the residual pattern is less strong.

- Figure caption 10: "small crosses mark station residuals mostly independent of variations in backazimuth"

Done

- Figure caption 10: "Tectonic map" -> "Tectonic units"; provide a proper reference for Handy 20XX if possible, otherwise Handy personal communication?

Added a reference to the MB4D website, where the material can be found.

- l. 421: -> "The resulting vertically stacked traveltime differences (...) between our crustal 3-D model and the crustal 1-D model of Diehl et al. 20XX can be found..." You should add something to this paragraph such as "Crustal contributions to traveltime residuals related to Ivrea body or sedimentary basins (Po etc) are in the order of XX seconds and comparable to the crustal corrections derived by

Waldhauser et al. 2002 and therefore significant when compared to the potential mantel signals and need to be removed…" Otherwise the aim/purpose of Figure A2 is not clear.

Done

- l. 423: -> "The most striking features of the stacked traveltime residuals after crustal correction are the…"

Done

- l. 428: "Defining a separate negative anomaly…" -> this sentence is odd and unclear and needs to be rephrased.

Done

- l. 432: This section and the following ones seem better suited for the discussion (see my general comments)…

Done, see above

- l. 437: "…move around…" -> rephrase! E.g. "location of anomalies varies depending on azimuth" or something?

Done

- l. 440: "… we imagine" -> "we consider"?

Obsolete

- l. 449: "with laterally moving imprint" -> Try to rephrase! "with locations depending on azimuth of incoming wavefield" or something?

Done

- l. 456: "… can not…" -> "cannot"

Done

- l. 463: "The high velocity anomaly building (C)…" -> meaning unclear. You mean the anomaly labeled "C"? rephrase

Done

- l. 466: How are these outlines defined (see my general?

Done, see above

- l. 472: "Up to now not much was said about…" odd, please rephrase…

Done

- l. 474: "using the techniques described above" -> "using the same procedures as for the high-frequency data (including azimuthal binning, crustal corrections, etc.)" Then you can remove the repetition around line 489 and 490.

Done

- l. 475: "We find that the obtained maps differ systematically…" -> Add a reference to Figure 12 showing the residuals for the low-frequency data. Otherwise it is not clear what you are refereeing to.

Done

- l. 492: "less strong" -> "smaller"? Can you quantify this? E.g. in terms of seconds or in %. Is the difference between the two datasets still significant compared to the mean errors or your picks?

Done

- l. 496 and elsewhere: "disparity" -> Not sure, if this is the right term. I would prefer "difference"

Disparity -> sign/difference

- l. 521: -> "Sophisticated, automatic single-channel picking approaches apparently do not achieve the targeted accuracy for teleseismic traveltimes."

Done

- l. 530: "These are absolutely essential" -> repetition, please delete sentence… "To overcome this problem…"

Done

- l. 531: -> "It proved to be sufficient…"

Done

- l. 539: -> "into the error estimation"

Done

- l. 539: -> "the higher …, the better"

Done

- l. 540: -> "The definition nicely includes…" -> try to rephrase

Done

- l. 544: -> "… conspicuous bumps should appear…" odd expression, rephrase!

Bumps -> deformations

- l. 553: what is "high-reaching fast upper mantle"? You mean "Upper mantle at shallow depths with fast seismic velocities"?

Obsolete

- l. 554: -> "The strike of the anomaly A correlates with the strike of the Apenninic mountain chain…"

Done

- l. 555: -> what is "slab-like" material? Why not "lithospheric slab"? I think there is evidence for its existence from other studies…

Done

- l. 584: -> "The AlpArray network proved to…"; "high correlation" in terms of what?

Clarified this

- Figure caption A2: Should be improved. If I understand correctly you show the difference between your updated 3-D crustal model (compiled from different crustal models) with the 1-D crustal model of Diehl et al 2009 for a vertical plane wave? It shows the potential crustal contribution to the teleseismic residuals which needs to be corrected for. This is not clear from the current caption.

Done

- l. 632: Somewhere here you might also acknowledge the contributions and comments of editors and reviewers of your manuscript…

That is definitely true!